# Recovery of logged forest fragments in a human-modified tropical landscape during the 2015-16 El Niño

Matheus Henrique Nunes[1,2✉], Tommaso Jucker[1,3], Terhi Riutta[4,5], Martin Svátek [6], Jakub Kvasnica[6], Martin Rejžek [6], Radim Matula[7], Noreen Majalap[8], Robert M. Ewers [4], Tom Swinfield[1], Rubén Valbuena [1,9], Nicholas R. Vaughn[10], Gregory P. Asner[10] & David A. Coomes [1✉]

The past 40 years in Southeast Asia have seen about 50% of lowland rainforests converted to oil palm and other plantations, and much of the remaining forest heavily logged. Little is known about how fragmentation influences recovery and whether climate change will hamper restoration. Here, we use repeat airborne LiDAR surveys spanning the hot and dry 2015-16 El Niño Southern Oscillation event to measure canopy height growth across 3,300 ha of regenerating tropical forests spanning a logging intensity gradient in Malaysian Borneo. We show that the drought led to increased leaf shedding and branch fall. Short forest, regenerating after heavy logging, continued to grow despite higher evaporative demand, except when it was located close to oil palm plantations. Edge effects from the plantations extended over 300 metres into the forests. Forest growth on hilltops and slopes was particularly impacted by the combination of fragmentation and drought, but even riparian forests located within 40 m of oil palm plantations lost canopy height during the drought. Our results suggest that small patches of logged forest within plantation landscapes will be slow to recover, particularly as ENSO events are becoming more frequent.

[1] Department of Plant Sciences and Conservation Research Institute, University of Cambridge, Cambridge CB2 3QZ, UK. [2] Department of Geosciences and Geography, University of Helsinki, Helsinki 00014, Finland. [3] School of Biological Sciences, University of Bristol, Bristol BS8 1TH, UK. [4] Department of Life Sciences, Imperial College London, Silwood Park Campus, Buckhurst Road, Ascot, Berkshire SL5 7PY, UK. [5] School of Geography and the Environment, Environmental Change Institute, University of Oxford, Oxford OX1 3QY, UK. [6] Department of Forest Botany, Dendrology and Geobiocoenology, Faculty of Forestry and Wood Technology, Mendel University in Brno, 613 00 Brno, Czech Republic. [7] Department of Forest Ecology, Faculty of Forestry and Wood Sciences, Czech University of Life Sciences Prague, Kamýcká 129, Prague 165 00, Czech Republic. [8] Sabah Forestry Department, Sandakan 90009, Malaysia. [9] School of Natural Sciences, Bangor University, Gwynedd LL57 2UW, UK. [10] Center for Global Discovery and Conservation Science, Arizona State University, Tempe AZ and Hilo, Tempe, HI, USA. ✉email: matheus.nunes@helsinki.fi; dac18@cam.ac.uk

Natural forest regrowth could offset 25% of current annual fossil fuel emissions (10 GtC year$^{-1}$)[1,2], helping to stabilise atmospheric $CO_2$ concentrations as we transition to a low-fossil-fuel economy in the coming decades[3]. Natural recovery of secondary forests in tropical regions is highly variable[4,5], partly because the remnant patches are often highly fragmented, creating heterogeneous environmental conditions[6]. Whilst advances have been made in understanding the effects of fragmentation on old-growth forests[7], much less is known about how fragmentation impacts regenerating forests[8]. The effects may be quite different: old-growth forests respond to fragmentation by losing biomass close to edges and shifting toward early successional stages[7], while fragmentation processes and secondary succession occur simultaneously in the regenerating forests[9]. Understanding the effects of fragmentation on secondary forest growth processes is thus vital for predicting the carbon benefits of protecting tropical forests, given that more than 50% of all tropical forests are degraded and increasingly fragmented[10].

The effects of fragmentation have been amplified by dry conditions experienced during El Niño Southern Oscillation (ENSO) events[11], which are becoming more frequent due to climate change[12–14]. Large uncertainty remains regarding the responses of regenerating logged forests to climate change[15], particularly because rising $CO_2$ concentrations are expected to increase biomass growth of degraded forests[16] and drought resistance differs among tree species[15]. Early successional species associated with recovering logged forests have characteristics that allow them to grow fast when water is plentiful[17–19]. These characteristics include large vessel diameters and high maximum stomatal conductances, which are associated with high transpiration and assimilation rates when the water supply is sufficient[20] but can make them susceptible to drought[21]. Increased potential evapotranspiration driven by high vapour pressure deficits (VPD) alongside reduced rainfall makes conditions difficult for plant growth[22]. Physiological responses to water shortage, include stomatal regulation, leaf shedding and reduced leaf production[23], with species varying enormously in their ability to resist drought[15]. The fragmentation effects are exacerbated by drought events in old-growth forests[11], but little is known about these climatic effects on recovering logged forests.

The position of fragments within plantation landscapes could also be a factor in recovery processes. Oil palm companies that have joined the roundtable for sustainable oil palm are committed to the protection of forests along rivers and on steep slopes within their estates. Differences in the geomorphic location of these patches could be important for recovery, because topography modulates soil and air moisture conditions, thereby impacting population-level responses to drought[24,25]. There is currently no consensus on how landscape position affects drought response, with both higher[26] and lower[24] mortality rates reported for wetter areas. Differences in forest composition may be important here because species differ greatly in their drought tolerance[27], indicated by differences in leaf turgor loss point[28], wood density, vessel hydraulic diameter, vessel area, stem cross-sectional sapwood[29] and capacity to refill cavitated xylem[30]. Despite the urgent need of research on tropical riparian buffers[31,32], variation in canopy dynamics between hills and riparian areas has not been adequately explored because the complexities of the fragmented landscapes are difficult to sample effectively in the field.

Repeat LiDAR surveying is a powerful technique for monitoring forest dynamics over large spatial scales, capable of providing fresh insights into the interacting factors affecting dynamics[33], and is increasingly used to model structural changes in forest canopies[34–36], and to detect tree mortality and growth, as well as leaf shedding and branch fall events[35–37]. Current knowledge of tropical forest canopies' responses to climatic variation derives from networks of permanent forest inventory plots spread across the tropics[38]. Whilst these plots provide many valuable insights, they sample <0.001% of the tropical landmass, and provide little information on spatial variation in forest response to drought, with degraded forests grossly underrepresented. The heterogeneous structure and composition of tropical forests make it challenging to infer change from small numbers of forest inventory plots[27], and remote sensing technologies, such as LiDAR, provide a way to measure changes in forest canopies across gradients of topography, fragmentation and degradation[39,40].

Here we investigate the relative canopy growth rates of human-modified tropical forests in Borneo using LiDAR surveys that spanned the 2015–2016 ENSO event. We estimate canopy height change within 36,655 $30 \times 30$ m pixels from repeated LiDAR surveys over an area of ~3300 ha of logged forests and use high-resolution topographic data and canopy height models (CHMs) as a natural experiment to examine the immediate influence of the extreme climatic events on forest canopies. Using a combination of LiDAR surveys, continuous topographic models, field measurements of the canopy, tree growth and mortality as well as measurements of microclimatic variables, we test whether: (1) non-fragmented regenerating logged forests had positive canopy height growth during the 2015–2016 ENSO; (2) fragmentation had negative effects on the canopy height growth of regenerating logged forests, as edge effects are likely to amplify tree mortality and reduce tree growth; (3) forest growth in low-lying areas was less affected by the ENSO event than on slopes and hilltops, where access to soil water is expected to be lower.

We find that regenerating logged forests continued to grow during the 2015–2026 ENSO event, despite the high temperatures and VPD in logged forests. However, we show that fragmentation effects increased exponentially with proximity to oil palm plantations and that forest canopies on hilltops were more affected compared to those in valleys and riparian areas. Our analysis demonstrates that these negative ENSO effects were the result of increased leaf shedding, canopy dieback, tree mortality and reduced tree growth in fragmented patches of regenerating logged forests, especially those left on hilltops.

## Results

**Climatic trends and vegetation responses to ENSO measured in field plots.** The ENSO event of 2015/2016 was relatively weak across Borneo[41], but VPDs linked to high temperatures may still create periods of high evaporative demand[22]. Eastern Sabah in Malaysian Borneo typically has an aseasonal climate. However, a running 30-day precipitation time-series from daily precipitation measurements in the stability of altered forest ecosystems (SAFE) Project in Sabah shows decreased precipitation between January 2015 and April 2016 linked to ENSO, with average monthly rainfall of $169 \pm 61$ mm compared with the long-term average of $235 \pm 61$ mm (Fig. 1a). Local average monthly temperature and average monthly VPD, or atmospheric dryness, measurements, obtained from the understory of 129 permanent forest plots between June 2013 and October 2018 indicate particularly high temperatures and VPD in March 2016 in the forest understory, one month before the second LiDAR survey took place, with the highest temperatures and VPD during the El Niño (March 2016) exceeding the long-term average of non-El Niño years (2013–2014) by 2.1 °C and 140%, respectively (Fig. 1b, c).

The long-term field-estimated top-of-canopy height (TCH) growth between January 2013 and November 2014 from 38 permanent plots revealed a pre-ENSO growth of 0.62 m per year (95% confidence intervals (CI): −0.76 to 2.01 m per year). During the ENSO years, the canopy continued to grow 0.3 m per year

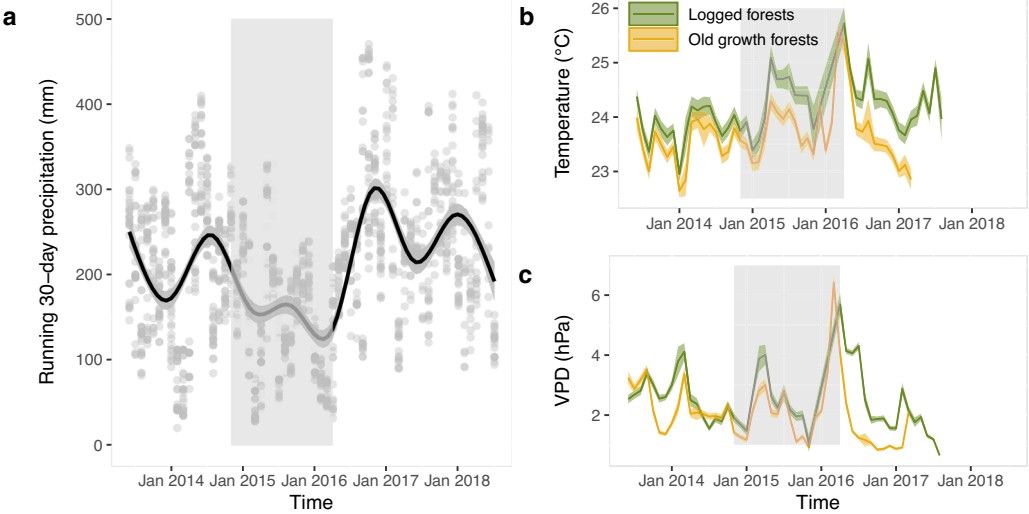

**Fig. 1 Reduced precipitation and increased temperatures and vapour pressure deficit during the 2015–2016 ENSO event.** Changes in climatic variables between 2013 and 2018 with the period between both LiDAR surveys (November 2014–April 2016) which were coincident with the ENSO event highlighted in light grey. **a** The running 30-day precipitation at the SAFE project in Northern Malaysian Borneo (mm) with 2191 grey points showing daily values in each month and a solid black line representing a cubic smoothing spline with 95% confidence intervals as the shaded area. **b** Mean daily local air temperature (in °C) and **c** mean daily vapour pressure deficit (VPD) computed from 939,388 relative humidity (RH, in %) measurements recorded within 129 permanent forest plots across old-growth and logged forests with shaded areas representing the 95% confidence intervals. VPD is the difference between how much moisture the air can hold before becoming saturated and the amount of moisture present in the air.

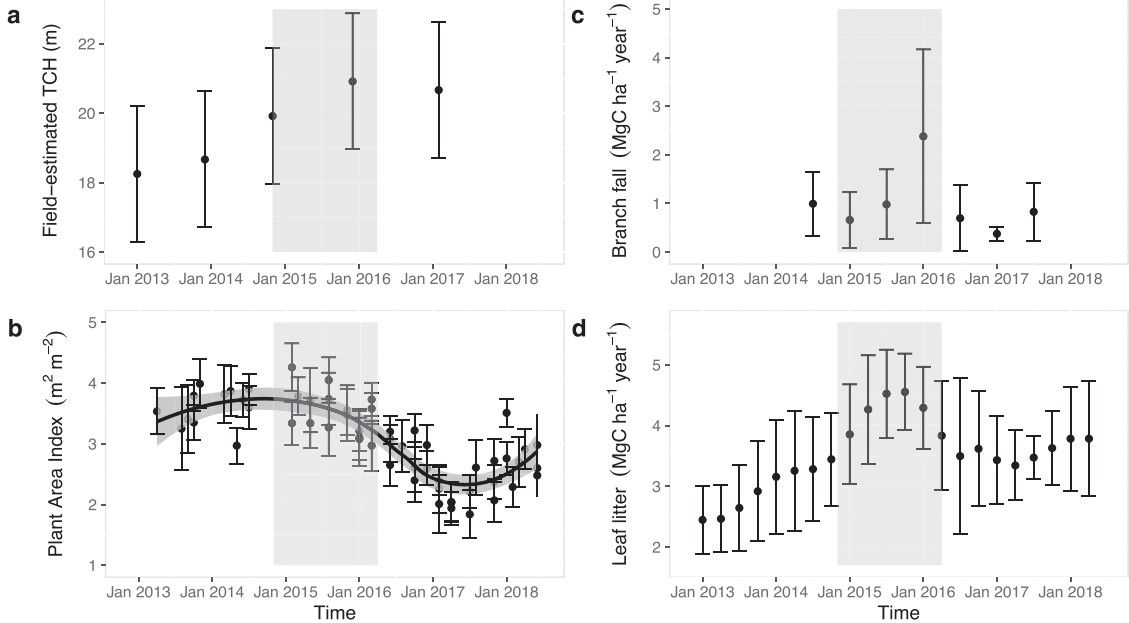

**Fig. 2 The 2015–2016 ENSO effects on canopy properties measured in the field.** Time series of **a** top-of-canopy height (TCH), **b** plant area index, **c** branch fall and **d** mean leaf fall measured in forest inventory plots between 2013 and 2018. The period between LiDAR surveys, shown in grey, coincided with the ENSO event. Means with 95% confidence intervals are shown.

(95% CI: −0.08 to 0.68 m per year) in 2015 but it fell to -0.06 m per year (95% CI: −0.50 to 0.37 m per year) in 2016 (Fig. 2a), despite the non-significant variations in canopy height growth ($t = -1.24$; $P$ value $= 0.21$).

The plant area index (PAI) time-series demonstrates sharp declines late in the ENSO that continued until January 2017, with PAI remaining low for another whole year (Fig. 2b). The long term pre-El Niño (August 2013–January 2015) average PAI of 3.6 $m^2 m^{-2}$ (95% CI: 3.2–4.0 $m^2 m^{-2}$) decreased to an average PAI of 3.3 $m^2 m^{-2}$ (95% CI: 3.1–3.7 $m^2 m^{-2}$) in March 2016, when temperatures and VPD reached their long-term

highest values (Fig. 1b, c). Despite the increased local precipitation after the highest temperatures and VPD in the late El Niño, PAI values continued to decline for the following ~10 months to 2.6 $m^2 m^{-2}$ (95% CI: 2.3–2.9 $m^2 m^{-2}$) until January 2017. The recovery in PAI started to happen only in January 2018, ~20 months after the peaks in evaporative demand brought about by the El Niño. The PAI decrease during the El Niño was coincident with an increase in branch fall early 2016 ($F$-value $= 3.14$, $P$ value $= 0.0337$; Fig. 2c) and in leaf litter throughout 2015 ($F$-value $= 44.96$, $P$ value $< 0.0001$; Fig. 2d).

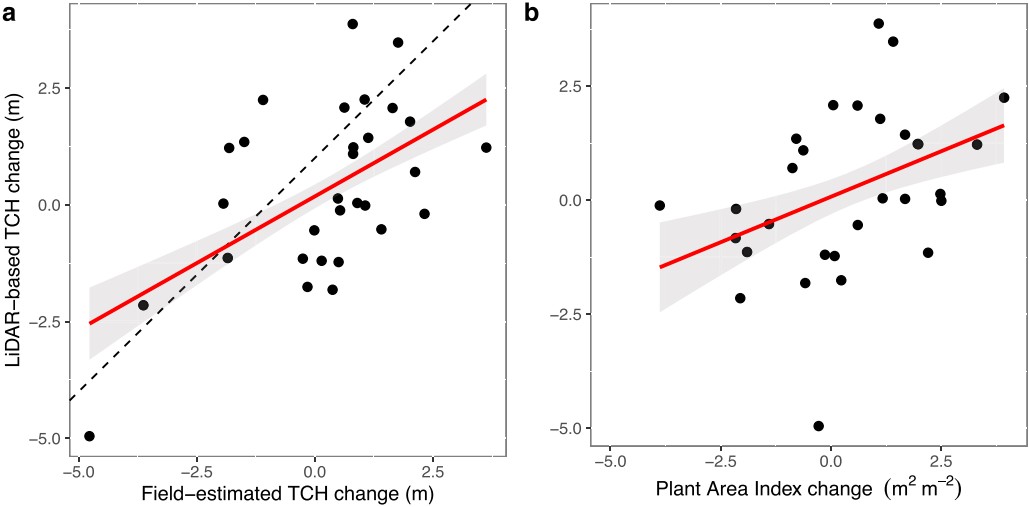

**Fig. 3 Correlation between LiDAR-derived and field-estimated change in canopy structure during the 2016–2015 ENSO event.** LiDAR-based top-of-canopy height (TCH) change (m) versus **a** field-estimated TCH change (m) and **b** plant area index (PAI) change ($m^2 m^{-2}$) between November 2014 and February 2017 from 38 permanent forest inventory plots (SAFE plots, each 25 × 25 m in size). Black dots represent the permanent plots (and missing plots are due to the presence of clouds during the first and/or second flights). The red lines represent values predicted by multiple linear regression, with the shaded grey area depicting the 95% confidence intervals.

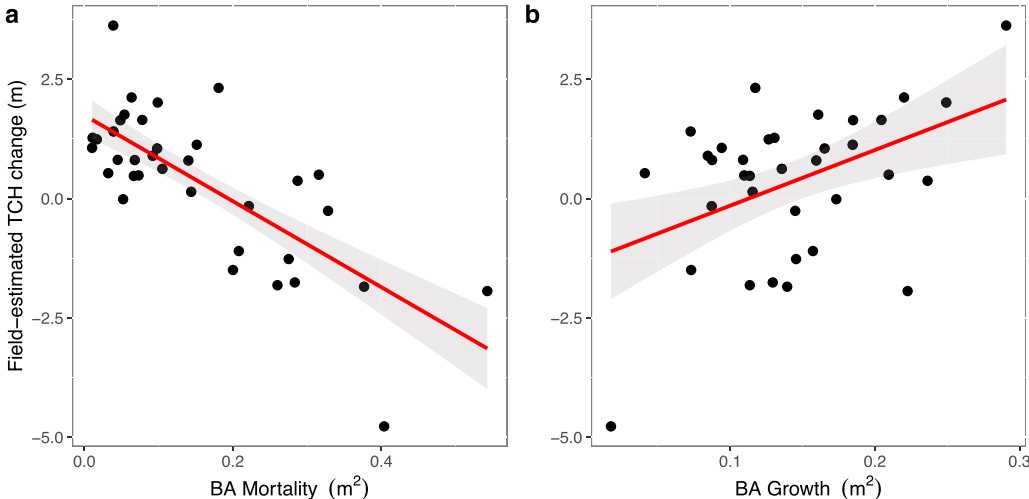

**Fig. 4 Correlation between canopy height and forest plot dynamics rates.** The effects of **a** basal area (BA) loss arising from stem mortality and **b** basal area growth on the field-estimated top-of-canopy height from 38 permanent forest inventory plots (SAFE plots, each 25 × 25 m in size). Black dots represent the permanent plots (and missing plots are due to the presence of clouds during the first and/or second flights). The red lines represent predicted values from multiple linear regression (i.e. field-estimated TCH change $= \beta_0 + \beta_1$ BA mortality $+ \beta_2$ BA growth) with 95% confidence intervals shown in grey.

**Correlations between LiDAR and field surveys of TCH and PAI changes.** Field-estimated TCH changes during the ENSO event from November 2014 to February 2017 of 0.23 m (95% CI: −0.30 to 0.77 m) were of similar magnitude to the LiDAR-based TCH change of 0.21 m (95% CI: −0.49 to 0.91 m). We found that 43% of the LiDAR-based TCH change, as demonstrated in Fig. 3a, b, is explained by a combination of changes in PAI (F-value = 8.8; P value = 0.0061) and changes in the field-estimated plot height (F-value = 10.9; P value = 0.0027), which represents changes in tree canopy density and basal area caused by mortality and growth of surviving trees (Fig. 4a, b). A multiple linear models shows significant effects of mortality (F-value = 78.9; P value < 0.001) and basal area growth of the surviving trees (F-value = 33.7; P value < 0.001) on the field-estimated canopy

height, explaining 77% of the total variation in field-estimated TCH change (Fig. 4).

Linear models of field-estimated canopy height change from 38 permanent plots were not able to detect statistically significant effects of TCH$_{2014}$ (F-value = 1.58, P value = 0.21), TPI (F-value = 0.27, P value = 0.60) or distance from oil palm plantations (F-value = 0.00, P value = 0.96).

We also investigated the relationship between aboveground biomass (AGB) and TCH (Supplementary Fig. 11). Despite the high correlation between TCH change and AGB change (F-value = 30.4, P value < 0.0001; Supplementary Fig. 12), the relationship between TCH and AGB was not strong enough to capture changes in AGB robustly and hence we opted to predict canopy height change across the landscape.

**Landscape drivers of canopy height change during the 2015/2016 ENSO**. The repeated high-resolution LiDAR surveys covered 3300 ha of a forest—oil palm landscape with forest canopy heights ($TCH_{2014}$) varying from 0 to 64 m, with distance from the forest interior to the oil palm plantation edges varying from 0 to nearly 4200 m. The undulating landscape had topographic position indices (TPI) varying from −24 (rivers and deep valleys) to 35.1 (elevated hilltops) (Supplementary Fig. 5).

Environmental conditions exerted key roles in the vegetation dynamics during the climatic event. The most parsimonious model to predict changes in canopy height (ΔTCH) included the effects of TPI and $TCH_{2014}$, as well as an asymptotic term describing the diminishing effect of distance from the edge (Eq. 1; AIC = 15,709; Supplementary Table 1):

$$\Delta TCH_i = 0.6576 - 0.02142\,TPI_i - 0.0172\,TCH_{2014i} \\ - 0.8553\,e^{-0.0049\,D_{Edgei}} + \varepsilon_i. \quad (1)$$

Forests closer than 300 m from oil palm plantations, with average TPI and canopy height, exhibited canopy loss. Although the total variation in canopy height change due to fragmentation in forests at different successional stages is similar, we demonstrate that the effect of fragmentation on canopy height growth was less pronouced in regenerating logged forests than in mature forests. Canopy height change in regenerating logged forests varied from a 0.55 m increase (95% CI: 0.48–0.59 m) in the interior, to zero net change (CI: −0.23 to 0.19 m) at about 70 m from the edge and a 0.4 m decrease (CI: −0.22 to −0.61 m) when in close proximity to oil palm plantations. The canopy height reduction resulting from fragmentation was also impacted by the topographical effects, with forests on ridges more affected by ENSO than low-lying forests (i.e., in valleys and riparian forests). To illustrate the effects of fragmentation and topography on forest recovery during the 2015–2016 ENSO event, we predicted ΔTCH for forests with a canopy height of 5, 20, and 35 m growing on valleys (5th quantile TPI = −8.2) or hilltops (95th quantile TPI = 9.0) in relation to distance from oil palm plantations (Fig. 5). We also illustrate the pre-ENSO long-term growth from permanent plots and demonstrate that forest height growth of regenerating logged forests during the ENSO years was similar to the pre-ENSO long-term growth, whereas taller, mature forests tended to have higher positive growth in pre-ENSO years. The canopy of regenerating forests (5-m canopy height) situated in the low-lying positions of the landscape experienced height loss within 40 m of edges between two census periods because of increased leaf shedding, branch fall, tree mortality and decreased productivity. This loss due to fragmentation was pronounced on the hilltops, with canopy loss within 110 m of edges. A map of predicted ΔTCH, based on Eq. 1, shows variation across the studied landscape in relation to $TCH_{2014}$, TPI and distance from oil palm plantations (Supplementary Fig. 10).

## Discussion
Repeat high-density airborne LiDAR across the human-modified forests of Borneo provides a unique perspective on the environmental factors affecting forest growth during the 2015–2016 ENSO. We demonstrate that regenerating logged forests in this landscape—which contain a high abundance of pioneer tree species[17] with acquisitive traits[18]—continued to grow, despite high temperatures and VPDs. However, our models show that environmental conditions modulated regrowth at the landscape level. Fragmentation had a negative impact on the forest growth near oil palm plantations, which is consistent with studies reporting that long-term fragmentation leads to greater tree mortality and lower productivity of forest near edges[11,42]. In addition, we demonstrate that the position of fragmented forests across the landscape was also a predictor of forest growth, with valleys and riparian forests showing faster canopy growth than those on hilltops during the ENSO event. Despite the fundamental importance of forest plots in characterising long-term dynamics of tropical forests, the 38 permanent plots within our study area represented a too small sample size to capture the spatial variation in canopy growth caused by logging, topography and fragmentation by oil palm plantations. The study demonstrates the value of repeated airborne LiDAR surveys, that allow forest growth to be mapped over entire landscapes, providing a perspective on forest dynamics that cannot be achieved using forest plots alone.

Field-estimated canopy height based on basal area measurements made in 38 permanent plots provide a metric of vegetation structure. Our plot-based results indicated that the field-estimated canopy height growth of non-fragmented forests was unaffected by the ENSO. Previous plot and LiDAR-based assessments of canopy changes have shown short-term increases in mortality rates and decreased productivity due to droughts across a range of size classes in the Amazon[35]. Droughts can lead to immediate tree death because of hydraulic failure, or slower tree death up to two years later[43,44], if weakened trees succumb to the effects of storms, windthrows, pathogens and insect outbreaks[45]. Additional LiDAR surveys of the region are crucial to investigate whether a potential delayed drought-induced tree mortality affected the canopy.

Our long-term field observations demonstrate a sharp decrease in PAI towards the end of the ENSO event, with low PAI persisting for ~20 months after the peaks in temperature and VPD. PAI is a combination of leaf area index (LAI) and the area of woody components including trunks and branches[46], so PAI loss may be attributed to increased leaf shedding and/or branch fall during the mid to late ENSO period. Increased leaf litter in response to increased evaporative demand during ENSO has also been observed in old-growth forests of Borneo[47], and losses in LAI may also be linked to stunted leaf development due to low water availability[48]. An increase in branch fall in response to ENSO droughts has also been reported in one Amazonian study[35], but not in another[49]; more research is needed given that branch fall can contribute significantly to carbon fluxes in tropical forests[50].

While 43% of the variation in canopy growth was explained by changes in PAI and changes in field-estimated plot height, we were unable to explain a large amount (~57%) of the variation in canopy growth. This residual variance could be associated with varying species composition of the plots (like sensitivity to drought can vary considerably among species), as well as canopy variables that we were unable to quantify or adequately account for. These include the uncertainties in height and crown area predictions used to estimate field canopy height, as well as PAI estimations based on canopy openness measurements. In addition, LiDAR-based canopy heights were obtained using different sensor configurations and flight parameters, which can affect canopy height estimation[51]. To minimise this effect, we restricted our analysis to areas with high point density in the 2014 LiDAR survey. We also tested the sensitivity of our results to spatial variation in point density in the 2016 LiDAR survey but found no evidence that this affected our conclusions.

The LiDAR surveys show that regenerating forests - away from plantation edges - maintained positive height growth during the ENSO event, whereas taller forests had near-zero growth. This result is consistent with studies from the Neotropics showing that young secondary forests have relatively high growth rates[4], and with other studies focussing on recovery of logged forests[17,52,53]. These recovering forests tend to be dominated by pioneer species with acquisitive traits that maximise carbon capture and

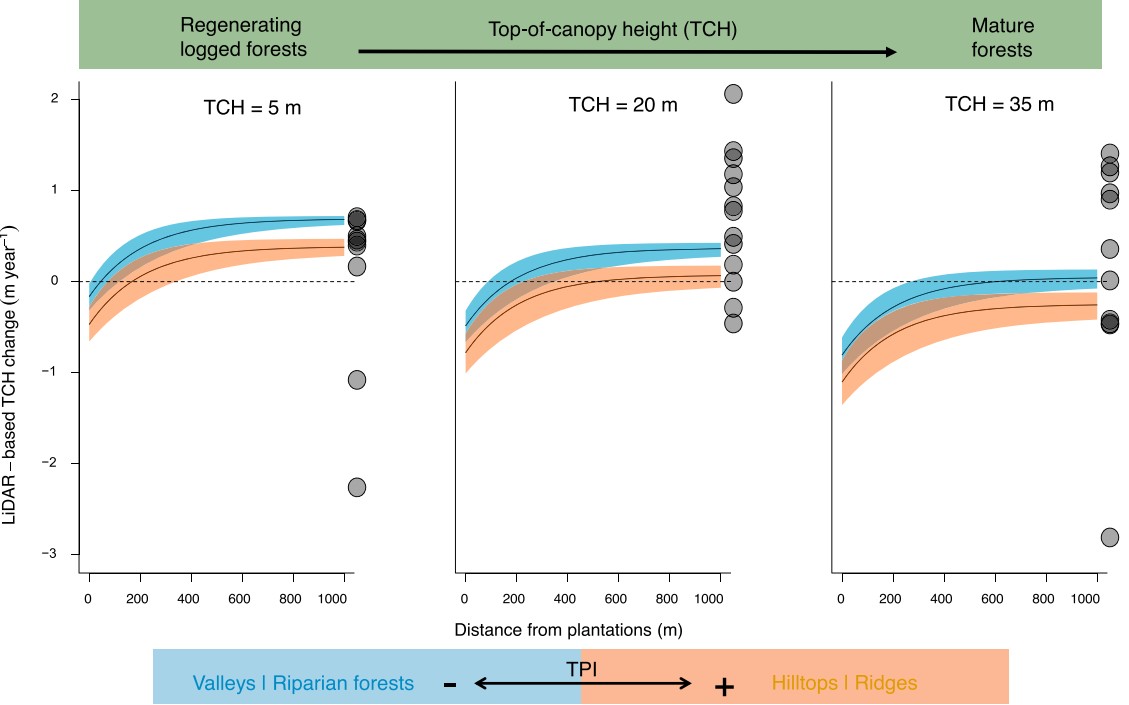

**Fig. 5 Fragmentation and topographic effects on canopy height change during the 2015–2016 ENSO event.** Predicted effects of distance from oil palm plantations and topographic position (TPI) on canopy height growth, obtained by fitting nonlinear models with a spatial autocorrelation structure. The solid black curves are predictions based on median parameter values obtained by fitting models to 24 subsets of the dataset, each composed of 5000 measurements; 95% confidence intervals are based on uncertainty in those parameter estimates (see Supplementary Methods 7 and 12 for details). Predictions are shown for short, medium and tall canopies, with initial heights of 5, 20, and 35 m.

growth[18]. The high abundance of pioneer species—which make up >50% of the total basal area of heavily logged forests in the SAFE experiment — can also make regenerating forests more vulnerable to higher temperatures and drought, as pioneer species tend to be less well suited to coping with the high evaporative demands and lower soil water availability that characterise ENSO events[21]. However, our results show that regenerating logged forests that were away from plantation edges continued to grow in height during the 2015–2016 ENSO event.

We observed strong edge effects (Fig. 5). Trees on forest edges tend to transpire more than those in forest interiors, depending on species-specific responses of stomata to water loss[54]. It is well known that old-growth forest near edges can have lower canopy height, which affects understory microclimate[55] and increases wind exposure[56], but less is known about the effects of fragmentation on forest recovery processes. The regenerating logged forests throughout the SAFE landscape had higher temperatures and VPDs in the forest understory, due to their low canopy height[57]. One possible explanation for the strong fragmentation effects on regenerating forests is that these fragments are exposed to increased wind speed and turbulence[56] which can lead to tree mortality and leaf shedding when the VPD of the air is high. Most fragmented landscapes are also altered by other anthropogenic changes, such as logging, hunting, fires, and pollution, which can interact synergistically with fragmentation[58], making it difficult to untangle the anthropogenic effects on forest structure.

The position of forests within the landscapes was also a strong predictor of canopy height growth during the 2015–2016 ENSO, with forests along rivers growing more rapidly than forests on hilltops. Our results suggest that topography contributes to landscape-scale variation in soil water availability, as well as water deficit and temperature of air, which in turn affect the degree of

dynamics rates associated with leaf shedding, decreased productivity and higher mortality within ENSO years. The topography is known to modulate species composition, functional diversity and vegetation dynamics[19,25,59]; forests growing on ridges may have strong competition for nutrients and water compared to those growing on deep alluvial soils or valley bottoms[60]. A previous study in Borneo reported that lowlying forests (i.e. swamps and riverine forests) dried out during the 1997–1998 ENSO drought, but were less water stressed than other forest types because groundwater was supplied from areas upslope, and tree mortality was low[24]. Fine-scale topographic variation amplified the impacts of the ENSO event, regardless of disturbance due to logging intensity and fragmentation, affecting forest dynamics at the landscape scale[61].

Oil palm companies that have joined the Roundtable for Sustainable Oil Palm are committed to the protection of high conservation value forests along rivers and on steep slopes within their estates. Riparian strips of native forest must be retained in plantation, primarily to reduce sediment loads entering water courses, although their importance for biodiversity and carbon cycling is increasingly recognised[31]. Our results suggest buffers have to be much wider than presently required - 20 metres on each side - in order to ensure that forest in the interior of the riparian strip continue to grow during droughts. If designed and protected appropriately, riparian reserves in oil palm estates support regrowth with potentially positive consequences for the global carbon cycle[62] and for ecosystem function[63]. Our results also demonstrate that small, fragmented patches of regenerating logged forests left on hilltops will be slow to recover due to lower water availability, particularly as ENSO events are becoming more frequent as a result of climate change[13,14]. Fragmentation in these regenerating logged forests was associated with canopy height loss

within 110 m of oil palm plantations, reflecting the intertwined effects of topographic position, fragmentation and climate. The higher exposure of these hilltops to erosion and weathering, which negatively influence soil properties and nutrient availability[64], may contribute to slow forest recovery and prove challenging for oil palm companies seeking to meet global standards of sustainable production by restoring natural forests in parts of their estates.

Given the rapid pace of land-use change across the tropics[62], the implications of this study extend beyond Borneo. With vapour-pressure deficits and temperatures predicted to increase through the 21st century in response to greenhouse-gas emissions, our results highlight the negative effects of forest fragmentation within oil palm landscapes during drought periods, particularly on small forest patches left on inaccessible slopes and hilltops. Our findings suggest that forests retained along watercourses will be less affected by droughts as they intensify in the coming decades, although we emphasise that different responses may be observed if forests experience greater water stress. The work highlights complex interacting effecs of climate change, topographic position and human disturbance on forests to regional warming and climatic variability. In light of the United Nations (UN) declaration that 2021–2030 is the Decade on Ecosystem Restoration, we voice concerns about the potential of heavily fragmented tropical forests to recover as climate becomes hotter and drier and highlight the need to protect riparian forests.

## Methods

**Study site**. The study is located in Sabah, Malaysian Borneo, within a region dominated by logged forests and oil palm plantations (4 38′ N to 4 46′ N, 116 57′ to 117 42′ E). Few regions have seen such rapid and extensive transformation as Borneo, where 163,000 km$^2$ (30%) of forest cover was lost between 1973 and 2010[65]. The study area encompasses the world's largest forest fragmentation experiment, the SAFE Project, located in an area gazetted for conversion to oil palm: as the plantations are established, patches of logged forests are being protected as part of an experiment seeking to establish the consequences of fragmentation on biodiversity and ecosystem functioning[66]. The SAFE Project site connects a 2200 ha block of mostly intact Virgin Jungle Reserve forest to a large area of degraded forest (over 1 million ha), most of which has been through one to three rotations of selective logging. Logging intensity has varied greatly over small scales due to differences in topography, proximity to roads and timber quality, which has created a complex mosaic of heavily to moderately logged sites[67,68]. At the community scale, logged forests contain a high proportion of pioneer tree species[17]. Heavily logged forests have 57.2% of the basal area represented by pioneer tree species, moderately logged forests 21.5% and slightly logged forests 6.9%, although recovering logged forests still retain old-growth forest species in low abundance. These pioneer species are characterised by acquisitive traits that maximise carbon capture and growth[18]. The forest modification gradient reproduces the real-world pattern of habitat conversion in Borneo, ensuring that phenomena observed in the study should be directly pertinent to policy issues in the region.

### Precipitation and microclimate time-series

*Precipitation*. Daily precipitation was systematically recorded from June 2013 to October 2018 at a weather station at the SAFE field station, which is within the region that was surveyed by LiDAR, and used to derive time-series of meteorological variables during the study period.

*Microclimate*. Air temperature (T, in °C) and relative humidity (RH, in %) were measured across a network of 129 permanent forest plots (each 25 × 25 m in size) established through the SAFE Project landscape[57]. These plots also include all the plots where tree measurements were conducted. Suspended Hygrochron iButton loggers (Maxim Integrated, USA) at a height of 1.5 m above the ground and shielded from direct solar radiation were used to record hourly T and RH readings in each plot (accurate to ±0.5 °C and ±5 %, respectively). Microclimate data were collected between May 2013 and August 2017, resulting in a total of 939,388 coupled T and RH readings. From the hourly temperature records, we calculated mean monthly temperature ($T_{\mathrm{mean}}$), directly related to biological activity across a range of taxonomic groups of tropical forests[69]. We used the microclimate data to characterise atmospheric water balance by estimating vapour-pressure deficit (VPD, in hPa). VPD is the difference between the saturation water vapour pressure ($e_s$) and the actual water vapour pressure ($e$) or atmospheric dryness. Given that $RH = \frac{e}{e_s} \times 100$, VPD can be expressed as $\left[\left(\frac{100-RH}{100}\right)\right] \times e_s$, where $e_s$ is derived from T using Bolton's equation[70]: $e_s = 6.112 \times e^{\frac{17.67 \times T}{T+243.5}}$. Having estimated VPD for each

coupled hourly observation of T and RH, we then calculated mean monthly VPD ($VPD_{\mathrm{mean}}$) for each study plot.

**Airborne laser scanning: data acquisition, fusion and height change estimation**. The first LiDAR data were acquired in November 2014 using a Leica ALS50-II LiDAR sensor flown by NERC's Airborne Research Facility. The sensor emitted pulses at a frequency of 120 kHz, had a field of view of 12° and a footprint of about 40 cm, with a mean point (i.e., return) density of about 13.2 points m$^{-2}$ (±13.2 points m$^{-2}$ standard deviation). The second LiDAR survey was conducted by the ASU Global Airborne Observatory (GAO; formerly the Carnegie Airborne Observatory[71]) in April 2016. The GAO LiDAR system was set to a combined-channel pulse frequency of 200 kHz, a field of view of 34° and a footprint of about 1.8 m, yielding a mean point density of 4.1 points m$^{-2}$ (±2.2 points m$^{-2}$ standard deviation). Despite the lower point density, GAO uses higher-wattage lasers with a larger beam divergence (0.5 mrad versus 0.22 mrad in the NERC data), which together result in an increased chance of ground detection in thick tropical understories. Further details of these flights can be found in Jucker and colleagues[72] and Asner and colleagues[40], respectively.

We used the LAStools (rapidlasso, GmbH; Gilching, Germany) suite of computational tools to process the data. To minimise errors in the fusion of both datasets, we first created a common digital terrain model (DTM) at 1 m resolution using a combination of ground returns from both surveys. For the NERC and GAO surveys, we identified ground points and interpolated ground and upper canopy returns into 2 m resolution maps of bare-earth ground elevation (DEM) and TCH above ground. Using a triangulated irregular network algorithm ("lasground"), a DTM was constructed from LiDAR ground returns. The heights above ground of all other returns were calculated by subtracting the DTM from their elevations. A CHM representing the upper height of vegetation was generated separately for each survey, following the methodology of Khosravipour and colleagues[73], and TCH above ground was initially calculated as the mean CHM at 2 m resolution using the LAStools (rapidlasso, GmbH; Gilching, Germany) suite of computational tools to identify ground points and interpolate ground and upper canopy returns for each flight line. It has been demonstrated that TCH, as measured by LiDAR, is a useful metric for estimating structural attributes of natural tropical forests and is relatively insensitive to sensor and flight specifications[74]. From the two LiDAR measurements of TCH obtained in 2014 and 2016, namely hereafter as TCH before ($TCH_{2014}$) and TCH after ($TCH_{2016}$) the ENSO, TCH change (or ΔTCH) at each 30 × 30 m cell over the wider SAFE landscape was calculated as $TCH_{2016} - TCH_{2014}$. The TCH change map was then coarsened from 2 to 30 m resolution following Asner et al.[40], considering the substantial reduction in uncertainties of TCH change estimates for lower resolution pixels owing to artefacts of repeat LiDAR data such as wind direction and within-canopy variation (Supplementary Fig. 1a, b).

The total area with repeated LiDAR flights covered 24,120 ha of forest and oil palm plantation mosaic. To extract regenerating logged forest pixels from the dataset, we excluded 9587 ha of oil palm plantations and 2500 ha of forest that was clear-cut ("salvage logged") between the two LiDAR surveys. Loss of biomass with logging can be due to the immediate damage caused by felling the selected trees, incidental damage to surrounding trees caused by the felled trees, and the infrastructure built for removing the logs out of the forest[75]. To avoid potential effects of logging that occurred in the interim of both flights on the surrounding forests due to infrastructure, pixels within 200 m of the clear-cut areas were also removed. Roads and their adjacent areas within 30 m were also removed due to their intrinsic differences in land cover compared to forest canopies. No other land-use types remained within the resulting study area. Finally, since LiDAR estimates can be affected by point density[76], biases arising from differences in point density were removed from the dataset. We demonstrate that an underestimation of tree height associated with point density <10 points m$^{-2}$ in the NERC dataset may have contributed to an overestimation of TCH change (Supplementary Fig. 2) and, therefore, we removed these pixels (~62% of the dataset). We also assessed the influence of point density variation in the GAO data and did not find any influence of point density of TCH change estimation (Supplementary Figs. 3 and 4). A small number of outliers that may have resulted from anomalies in the processing of the DTM and TCH or small misalignments were still detected, and thus we trimmed the lower and upper 1% of all TCH change values with the intention to eliminate unrealistic values. The final area analysed was 3301 ha (36,655 pixels) with $TCH_{2014}$ varying from 0 to 64 m (Supplementary Fig. 5a).

**Mapping topography and distances from the edge**. SAFE has a varied topography, with the lowlands (100–350 m a.s.l.) almost entirely converted to oil palm and the remaining forest predominantly covering hills rising to over 1000 m a.s.l. Most streams within oil palm plantations in the SAFE landscape have a forest buffer zone, the width of which ranges from 12 to 100 m, although some have no buffer[77]. A map of topographic position index (TPI) was generated from the combined LiDAR-derived DTM. TPI describes the height of a pixel relative to the surrounding landscape and ranges from negative where the terrain is concave (i.e., valleys) to positive where it is convex (i.e., ridges). TPI was calculated by first coarsening the resolution of the DTM to 10 m by spatial averaging, then mean TPI values were calculated within 1 ha neighbourhoods as in Jucker et al.[25]. The undulating landscape had TPI varying from −24 (rivers and deep valleys) to 35.1

(elevated hilltops) (Supplementary Fig. 5b). Based on a stream network of the SAFE Project, we show that low TPI values are generally associated with rivers (Supplementary Figs. 6 and 7a). Negative TPI values are likely to be within ~75 m from rivers and 50% of the forests with TPI < −5 are within 15 m from rivers; thereby the low-lying forests of our study generally represent riparian forests. To further understand how topographic position mediates water availability for trees, we used the DTM to calculate the topographic wetness index (TWI) for each 30 m grid cell using the 'dynatopmodel' R package[78]. High TWI values indicate grid cells with topographic characteristics favourable for accumulating higher levels of soil moisture, and vice versa[59]. The high correlation between TPI and TWI values across SAFE indicates that areas with more convex curvature and steeper slopes (higher TPI) tend to have lower TWI values, and vice versa (Supplementary Fig. 7b; Spearman correlation = −0.81, P value < 0.0001). Moreover, recent work at SAFE has shown that variation in TPI across this landscape is strongly related to variation in microclimate and nutrients[19,57].

Satellite imagery was used to classify land use, and calculate distance from forest to oil palm edge. Earth Imaging data from the Pléiades satellite constellation (EADS Astrium), acquired over the SAFE landscape in June 2016, were classified visually to define boundaries between forest and plantations using the software QGIS 3.2.3. Pléiades data comprise a 0.5-m resolution panchromatic band, and four spectral bands (blue, green, red, and near infra-red) with a 2.8 m spatial resolution, resampled to 2 m. The panchromatic band has sufficiently high resolution to distinguish forests, oil palm plantations and clear-cut logging. Oil palm plantations did not expand between the two flight times. The distance of each pixel from within the forest to the oil palm edge boundaries ($D_{Edge}$) was calculated using the gDistance function from the "rgeos" package in R, with values ranging from 0 to 4200 m (mean = 1825 m) (Supplementary Fig. 5c).

## Using field data to elucidate the mechanisms driving changes in canopy height.
Canopy height loss recorded by LiDAR could be a response to leaf loss, branch loss or tree death, while height gain could arise from leaf gain or upward stem growth. We used canopy height, PAI, branch fall and leaf litter from two separate permanent plot networks within the SAFE landscape to investigate how the 2015–2016 ENSO event affected the canopy and which factors were driving the changes observed by LiDAR. More specifically, we used tree mensuration data and canopy openness measurements from 38 "SAFE plots" established in 2011 (each 25 × 25 m in size), and PAI, branch fall and leaf litter measurements from eight 1-ha Global Ecosystems Monitoring (GEM) plots. Although these estimates cannot be used to directly validate the LiDAR measurements, they help us investigate the likely mechanisms driving canopy changes during the ENSO event.

## Canopy height growth.
We first used the SAFE plots to assess which factors were driving the changes observed by LiDAR. Stem diameters at 1.3 m height (DBH), or immediately above buttresses, of all trees ≥10 cm DBH were measured in these plots in January 2013, December 2013, November 2014, December 2015 and February 2017. Trees that exhibited extreme diameter growth (i.e., DBH growth ≥5 cm yr$^{-1}$ or shrinkage ≥12 s, where s is the standard deviation of the DBH measurement error, $s = 0.9036 + 0.006214$ DBH) were assigned the mean expected growth rate of trees of the same size in the same plot[79]. In addition, we also measured tree height (H) and crown area (CA) for a total of 3248 trees in the eight 1-ha GEM plots. We then used these data to fit allometric equations to predict H and CA for all trees in the 38 SAFE plots (n = 10,393; Supplementary Fig. 8). H was measured using an Impulse 200 laser rangefinder (Laser Technology Inc., Colorado, USA), while CA was estimated from the ground-based mapping of the horizontal crown projection of each tree, using the Field-Map technology (IFER, Ltd., Jílové u Prahy, Czech Republic; for details of the technology, see Hédl and colleagues[80]. Plot centre coordinates used by[67] were used to geolocate each permanent plot.

We then calculated the crown-area weighted height of each plot, which is comparable to the top-of-the-canopy height measured by LiDAR[81], where the height of each individual tree is weighted by the horizontally projected crown area. Since the upper canopy surface measured by LiDAR consists primarily of the tallest dominant and codominant trees, weighting by the projected crown area minimises the influence of smaller stems. The height of each tree was weighted by its proportional contribution to the total crown area to calculate mean TCH[81]. We ranked the trees by crown area size and selected overstory trees from this ranked list. To do this, we calculated the cumulative sum of canopy areas going down the list from the largest tree, until we reached 625 m$^2$, which is the plot area. All other trees were excluded and we calculated the crown-area weighted height (hereafter field-estimated TCH) of the selected overstory trees only. The field-estimated TCH agreed well with the LiDAR-derived TCH for both LiDAR surveys (Supplementary Fig. 9).

For comparison with the TCH changes obtained from LiDAR, we then calculated field-estimated TCH change as the difference between the end (February 2017) and the beginning (November 2014) of the interval. Field-estimated TCH change between those dates had a closer relationship with LiDAR-based TCH change compared to field changes between November 2014 and December 2015. Pre-ENSO field-estimated TCH changes from tree measurements made in January 2013 and November 2014 were also used to evaluate the long-term growth during non-ENSO years.

## Plant Area Index.
Canopy openness measurements were made in the SAFE plots between November 2014 and February 2017. Hemispherical photographs were taken at 17 locations in each plot at two different heights (1 and 2 m) using Sigma 4.5 mm f/2.8 EX DC HSM circular fisheye lens. The photos were thresholded in ImageJ (version 1.51j8) using Auto Threshold and IsoData methods to create binary bmp files (i.e., sky = 0, vegetation = 1). The R package "cimesr" was used to calculate weighted canopy openness for zenith angles 60° and 90°, using 40 zenith bands and 150 sectors[82]. PAI was then estimated from canopy openness using the method of Kalacska et al.[83]. We then calculated PAI change as the difference between the end (February 2017) and the beginning (November 2014) of the interval.

We also used PAI data from the GEM plots continuously measured between August 2013 and June 2018 to create a PAI time-series. These PAI estimates were derived from hemispherical photos taken at 25 locations in each plot following a regular sampling grid. Three photos were taken at each point and these were repeated approximately every three months over the duration of the study (1327 photos in total). The photos were analysed with Hemisfer software, with 60° field-of-view, a calculation based on Thimonier et al.[84], clumping correction based on Chen and Cihlar[85], and non-linearity and slope correction based on Schleppi et al.[86]. The PAI values match those in Pfeifer et al.[67] estimated for the wider SAFE landscape.

## Leaf shedding and branch fall.
We measured branch fall (July 2014–July 2017) and litterfall (January 2013–June 2018) in the GEM plots within the SAFE landscape to create a branch fall and litterfall time-series. Branch fall was measured two to four times per year using 2 m × 2 m quadrats (n = 25 per plot), where the branches >2 cm diameter were collected, divided into five decay classes[87], dried, weighed and corrected for mass loss due to decay[17]. The first survey quantified stock, rather than the production of new material, and thus excluded from the data. In subsequent surveys, fallen branches from dead trees were excluded, as we were interested in the branch turnover term. Litterfall, of which leaf litter constituted 89% ± 1.4%, was monitored using litter traps of 0.5 m × 0.5 m at 1 m heigh (n = 25 per plot). The litter was collected every 14–30 days, dried at 70 °C and weighed.

## Landscape drivers of canopy height change during the 2015/2016 ENSO.
Change in ΔTCH was modelled as a function of TPI, the 2014 estimate of canopy height ($TCH_{2014}$) and distance from oil palm plantations ($D_{Edge}$). We compared linear models containing these variables with the non-linear model $y = a - be^{(-cx)}$, where a includes TPI, $TCH_{2014}$, as well as the interaction terms $TCH_{2014} \times TPI$ and $TCH_{2014} \times D_{Edge}$, and $-be^{(-cx)}$ as an asymptotic component that represents the saturation of ΔTCH with $D_{Edge}$, denoted by x in the model. An asymptotic component in the model is more ecologically meaningful to investigate the edge effects on forest dynamics[11]. The models were fitted using the nls function in R[88].

After comparing all the models using AIC (Supplementary Table 1), the following was selected:

$$\Delta TCH_i = \beta_0 + \beta_1 TPI_i + \beta_2 TCH_{2014i} + \beta_4 e^{-\beta_4 D_{Edgei}} + \varepsilon_i \qquad (2)$$

where $\beta_0$ to $\beta_4$ are the model parameters and $\varepsilon_i$ is the normally-distributed residual error.

Spatial autocorrelation results in an underestimation of the true uncertainty in the fitted parameter values[89]. We incorporated a spatial correlation structure in the model using the nlme function in R to estimate model coefficients by maximum likelihood estimation[90], and spatial autocorrelation in the residual error $\varepsilon$ was modelled using an exponential function using corExp(form = ~X + Y), where X and Y are the plot coordinates (Supplementary Table 2). Although nlme is a mixed-effects modelling function, we employed it to account for spatial autocorrelation and did not include any random variables. This analysis was restricted to 5000 randomly selected pixels because spatial modelling using nlme is memory and time-demanding.

To test whether a 5000-pixel subset is sufficient to estimate unbiased parameter values, we first ran 24 randomised permutations of the Eq. 2 with the spatial autocorrelation structure for randomly selected 3000-, 4000- and 5000-pixel subsets from the 36,655-pixel dataset. We then generated the mean and coefficient variation (CV %) of parameter values for the assessment of model stability with increasing subset sizes (Supplementary Table 2). We also ran 24 randomised permutations of the Eq. 2 with no spatial autocorrelation structure to investigate whether mean parameter values differed from parameter values when using the full dataset (Supplementary Table 3). Given (i) the considerably smaller CV for the 5000-pixel subset and (ii) the similar mean parameter values of twenty-four 5000-pixel subsets to the full dataset's parameters, we demonstrate the consistency of 5000-pixel subsets to predict canopy height change across the landscape. The function intervals in R was used to predict the median and 95% confidence intervals ($CI_{model}$), and then adjusted to the full dataset ($CI_{corrected}$) as in $CI_{corrected} = CI_{model} * (n/N)^{0.5}$, where n is the number of variables used to estimate the parameter values in the model (5000 pixels) and N the total number of pixels in our analysis (36,655 pixels).

**Reporting summary**. Further information on research design is available in the Nature Research Reporting Summary linked to this article.

## Data availability

Repeated canopy height data, topographic position index and distance of forests from oil palm plantations generated from repeat LiDAR surveys and analysed during the study have been deposited in the UK Centre for Ecology and Hydrology and made publicly available with the identifier https://doi.org/10.5285/534838c8-0e1f-4a04-a837-2e19a4e93797. Microclimate data across permanent plots are openly available online from https://doi.org/10.5281/zenodo.1441585.

## Code availability

There is no particular code or mathematical algorithm that is considered crucial to the conclusions. All relevant R-functions that were used are referred to in the Method section (see package vignettes for details).

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

## Acknowledgements

We thank Glen Reynolds for suggestions in the preliminary phase of the manuscript, Matthew Struebig for helpful comments and Sacha Khoury for help with image processing and analysis. We are grateful to Marion Pfeifer for providing us with the Pléiades imagery used to distinguish forested areas from oil palm and Sui Peng Heon for providing local precipitation data. MHN was supported by a PhD scholarship from the Conselho Nacional de Pesquisa e Desenvolvimento (CNPq, grant No. 201516/2014-4) from Brazil. D.A.C. and T.J. were supported by NERC's Human-modified Tropical Forests Programme (Biodiversity And Land-use Impacts on tropical ecosystem function project; grant No. NE/K016377/1). T.J. is funded by a NERC Independent Research Fellowship (grant No. NE/S01537X/1). M.S., J.K., and M.R. were funded through a grant from the Ministry of Education, Youth and Sports of the Czech Republic (grant number: INTER-TRANSFER LTT17017). We are grateful to the former NERC Airborne Research Facility for the LiDAR survey and NERC's data analysis node for completing the post-processing stages. The Global Airborne Observatory flights and data processing were supported by the UN Development Programme, the Avatar Alliance Foundation, the Roundtable on Sustainable Palm Oil, the World Wildlife Fund and the Rainforest Trust. The Global Airborne Observatory is made possible by grants and gifts from private foundations, visionary individuals, and Arizona State University. We thank the Sime Darby Foundation, the Sabah Foundation, Benta Wawasan and the Sabah Forestry Department for their support of the SAFE Project. We thank the South East Asia Rainforest Research Partnership for logistical support in the field, and Yayasan Sabah, Maliau Basin Management Committee, the State Secretary, Sabah Chief Minister's Departments, the Malaysian Economic Planning Unit and the Sabah Biodiversity Council for permission to conduct research in Sabah. Publishing fees were partially covered by the University of Helsinki (profit centre: H5101; WBS-number: 75101002).

## Author contributions

D.A.C. coordinated the NERC airborne survey and originally conceived of the project. G.A. led the GAO airborne campaign and LiDAR data collection. T.J., T.S., N.V., and R.V. processed the LiDAR data and assisted M.H.N. with data analyses. M.S., J.K., M.R., R.M., T.R., and R.M.E. contributed field data. M.H.N. wrote the paper under the guidance of D.A.C. and assistance of T.J. All authors contributed to the revision of the paper.

## Competing interests

The authors declare no competing interests.
