## [Peer Review File · Nature Communications]

Reviewer comments initial -

Reviewer #1 (Remarks to the Author):

This paper has great data and the statistics appear to be sound, but it is not always well-explained.

Firstly it is not always clear what sort of forest the authors are writing about, since in the Abstract it is called 'regenerating forest', and 'early-successional forest', both of which are ambiguous and could include everything from lightly logged forest to regrowth after cultivation. This is important because even triple-logged forest retains most old-growth species on site, so they eventually dominate regrowth, even if there is an initial flush of pioneers, while regrowth after cultivation is dispersal-limited and dominated for decades by pioneers. If 'regenerating logged forest', used later, is correct, then this should be used throughout, including in the Abstract. Assuming this designation IS correct, it then seems unlikely to me that the canopy trees that dominate the results are all or even mostly the 'early successional species' whose anatomy and physiology favors drought intolerance. This should be checked in the permanent plots, but I think the implication that the regenerating forest canopy is dominated by pioneers is wrong.

Secondly, the Abstract implies that reductions in canopy height are due to leaf shedding and (whole tree?) mortality. Both should be obvious from the ground, while branch loss, mentioned later, is less clear from ground level. If you have evidence for branch loss, this should be mentioned in the Abstract and elsewhere, since it appears to an overlooked and neglected impact of droughts.

Thirdly, we are told that ENSO events are becoming more severe, but the evidence for this is still fairly weak. You need to give a balanced view on this, with support from the recent literature and CMIP6 projections.

Finally, I found the mixing of review and new results in the Discussion and Conclusions is often confusing. This section should primarily discuss YOUR results, and compare them with the literature where appropriate, but readers need to know what was learned from this study. A synthesis of all the evidence would also be useful, but it should come after the results of this study have been discussed.

Richard Corlett

Reviewer #2 (Remarks to the Author):

Review on manuscript "Recovery of logged forest fragments in a human-modified tropical landscape"

General comments:

The authors present data on forest regeneration after the 2015-16 ENSO event based on canopy height growth across a converted tropical landscapes in Malaysian Borneo using repeat airborne LiDAR surveys, field-based forest inventories and LAI measurements. Their main findings reveal that forest fragmentation and topographic position influence canopy growth, with hilltop forests and small forest patches affected more strongly and pervasively. The study further identified strong edge effects extending over 500 metres into early-successional forest having implications for forest management.

I am convinced that this study is a significant contribution to a better understanding of how climatic events such as reoccurring droughts will influence tropical landscapes in the future aiding guidelines for restoration and is thus of interest to the scientific community and the broader public. While some of the findings discussed in this paper are not completely novel and have been observed before based on field-based inventories, the spatial scope and detail on the interface between remote sensing and forest monitoring are not present for Southeast Asia and rarely across the tropics to my knowledge.

Overall, I find the study to be well written and comprehensible. To my understanding the methods and statistical analyses are valid and described in sufficient detail. My only principle critique regards the terminology of riparian forest in this study, which did not become sufficiently clear

from my point of view. Assumably they are being opposed to hilltop forests and roughly follow streamlines, but not every non-hilltop forest is a riparian forest in many tropical landscapes. It may be imaginable in a very rugged terrain, but then the authors should include a more information on the relief and altitudinal amplitudes in the study area. Overlaying the topographic position index with the stream network may not be a sufficient determinant as the soil water regime and consecutive effects on water deficit in trees is likely to be quite different in the areas labelled as riparian forests.

The conclusion and last paragraphs of the discussion are very insightful, but perhaps too detailed, slightly repetitive and at times a bit far-stretched particularly regarding the importance of riparian buffer zones given that soil water deficits have not been measured and the classification of riparian forest remains not concrete. Parts of the conclusion appear more like an introduction to physiological mechanisms behind the observed patterns or explanations that seems better placed at the beginning of the manuscript.

Specific comments:

Unfortunately, there are no line and page numbers in the pdf of the manuscript so I will copy the passage I'm referring to here and upload a file where the passages are marked. I hope it will be comprehensible this way.

Abstract

- Perhaps the study title could include a brief mention on the type of recovery investigated i.e. after ENSO event/ post-drought or similar.
- First sentence: I find using the term "loss of nature" is a bit unassertive and generalized as actually habitat transformation is taking place where it is possible to argue nature still exists. Maybe using natural habitat, pristine environment or primary forest would make it clearer.
- Sentence 6: Canopy height decrease seems not a direct logical cause of reduced growth. Maybe the authors can add stagnation or rephrase the sentence slightly.
- Sentence 8: Should it read: will be slow to recover even if protected"? Or will they recover faster if not protected?

Introduction

- Line3: CO₂ in subscript

Material and methods

Mapping topography and distances from edge

- In the investigated study area there were no other land-use types than oil palm plantations, open clear-cut area and forests present? Such as tree plantations or other agricultural use?
- The definition of riparian forests remained unclear to me. Did these areas have a certain distance to rivers or other water bodies? Where they periodically flooded? How much percentage of the forests with TPI < -5 is in close proximity to a river or stream? In these areas likely not only water supply but also soil nutrient supply may vary.

Field data for validation and mechanism elucidation

- For how many trees within the permanent the height and crown extent was measured? Was a rangefinder or Vertex used for the height measurements?
- "The field-estimated canopy height agreed well with the LiDAR-based canopy height for both LiDAR surveys." This could be specified a bit giving a measure of variance or model fit.
- "...LAI change were estimated as the difference between the end (February 2017) and the beginning (November 2014)." There is no expected seasonal influence of rainy or drier season on LAI in the study area? Are there deciduous or brevideciduous species present?
- How often were the hemispherical photos taken for the continuous measurements in the GEM plots?
- Figure 1 Header: Perhaps better to use "climatic variables" instead of climate

Discussion

- Line 4: "...when temperatures exceeded 2.1 °C and VPD exceeded 140% of the local long-term average." This is not entirely clear to me. Which timeframe was counted? Did the maximal or average temperature/VPD exceeded the long-term average by 140% or was it 140% of this average?

- "Fragmentation effects increasing exponentially with proximity to oil palm plantations are consistent..."

Forest recovery in human modified tropical landscapes

- Perhaps rephrase to: This result supports the high growth rates observed in secondary forests in the Neotropics

- This statement as stand-alone should be used more cautiously as the study actually identified hydraulic traits to influence the probability of mortality among species and was conducted on a subset of 20-50 species from seasonally dry tropical forest. There is plenty of robust evidence from tropical rainforests that tree size and other plant functional traits are linked with vulnerability to drought and mortality. E.g. see Anderegg 2016, 2019, Aleixo 2019

Conclusion

- "Droughts can lead to immediate tree death because of hydraulic failure, or slower tree death because of carbon starvation or a combination of both mechanisms, and can be synergistically coupled to the effects of storms, windthrows, pathogens and insects" That seems to belong rather in the introduction.

Figures

- Figure 5: It would be good to include a legend for the hilltop and valley colour code. The figure already has a lot of white space which could perhaps be improved.

Supplementary Information:

- Figure S2: The axis label size should be increased

- Figure S5: The field estimated TCH measurements were conducted nearly one year after the LiDAR ones when the peak of the ENSO-event was occurring. Do the authors expect any influence of the

- Figure S6: I did not find the figure mentioned in the text. It's a pity as I find the results very exciting.

Reviewer #3 (Remarks to the Author):

I was asked to review the manuscript written by Nunes et al. This manuscript demonstrates with repeat lidar that forest recovery during ENSO varies spatially and gives new insights into fragmentation of regenerating forests. Whereas the methods themselves are mostly building on existing work and methods, the context and conclusions are important. I was happy to see that the authors also included some recommendations regarding forest (fragmentation) management. Overall, this was a well written manuscript, but I found some sections overly detailed, whereas other sections did not have enough (or confusing) details to fully understand the methods that were used, which is critical. I did enjoy the discussion and I thought that was excellent. I invite the authors to respond to my concerns.

My main concerns are related to the lidar analysis and the detail in the manuscript has not convinced me that these 2 datasets are interoperable.

1) What is the beam divergence (and exit diameter) of both sensors? The authors report the footprint (I assume at top of canopy or ground) of 40cm vs 180 cm. They then use lastools to generate a DTM with 2m resolution. However, if the beam divergence is significantly different (I found a beam div of 0.22 mrad for NERC and 0.5 mrad for GAO), then the sensor with the smaller beam divergence will be able to penetrate the canopy more easily, having a higher probability of actually detecting a ground return. The other way around, the sensor with the larger beam divergence might not always detect the actual ground, but some averaged values of elements near the ground. This is quite important as this is the baseline for tree height.

2) Why is the NERC reported point density 4pts/m²? In Jucker et al 2018

(<https://onlinelibrary.wiley.com/doi/epdf/10.1111/gcb.14415>), this dataset has 15.3pts/m²?

Jucker et al 2018 mentions 4 pts per laser pulse. This links back to my previous comment on data-operability as the NERC data might now have 4 times more points and smaller beam divergence

(to be confirmed by the authors). The authors also mention that "only data with point density above 10pts/m² were used"; but the reported point density for the 2016 flight is only 2pts/m²?
3) Figure S1A: Is the caption of that figure correct? I think the authors should include these uncertainties in the analysis and figures and introduce error bars when reporting lidar TCH change.

I am also bit concerned about the "validation", or at least the use of this term. Whereas lidar gives you more or less a direct estimate of height; how can an estimate through an allometry (field height) be serving as a validation? What essentially is measured in the field is mostly DBH, not height. At the very least, the uncertainty of these allometry height values should be taken into account. How are these field H measurements done? (There is a sentence "canopy height growth: canopy height was estimated from the forest plot dataset ..") How was this done/measured and how does this link to the H-DBH allometry mentioned in the previous paragraph?

I have a similar issue with the term LAI for validation. It appears that what the authors did measure was PAI. However, this makes it more interesting as the effect of forest recovery on the woody and leaf component of PAI might not be equal. The last sentence in the 2nd paragraph of "correlations between ... LAI change" suggest that the sharp drop in LAI in 2016 is due to high temperatures, but looking at fig2A and B this sharp decline might already happen earlier, but the loss of LAI/PAI during El Nino might have been partly compensated by tree growth (and therefore increase in the woody component).

Smaller comments:

- Intro, first sentence "could offset much of the ..". It would be good to have some number of % here. How long before we will see an effect of restoration; does growth of very young trees offset much of these carbon emissions?
- Intro, start second paragraph ".. effects may be amplified ..". But you only do your analysis in the ENSO period; what is the relevance of this suggestion?
- Intro, 3rd paragraph "Differences in forest composition .. tolerance". This very recent discussion in <https://www.nature.com/articles/s41467-020-17214-4> might be worth including here?
- Study site: It would be good to have map of the area (in the supplementary) with an overlay of the extent of the 2 lidar flights
- Precipitation: Is there only 1 weather station? How do you then get information of precipitation in hilltops vs low-lying areas. Your 3rd objective seem to indicate that water availability is a factor, so how was this extrapolated over the extent of the study area?
- "3000 randomly selected pixels.. memory demanding": How memory demanding? I am surprised this is still an issue when cloud computing and computing clusters are often available. The authors should at least provide a figure (in supplementary) that 3000 pixels is enough.
- Will (or is) the data and code publicly available?

Response to Reviewer Comments -

Reviewer #1 (Remarks to the Author):

This paper has great data and the statistics appear to be sound, but it is not always well-explained. Firstly, it is not always clear what sort of forest the authors are writing about, since in the Abstract it is called ‘regenerating forest’, and ‘early-successional forest’, both of which are ambiguous and could include everything from lightly logged forest to regrowth after cultivation. This is important because even triple-logged forest retains most old-growth species on site, so they eventually dominate regrowth, even if there is an initial flush of pioneers, while regrowth after cultivation is dispersal-limited and dominated for decades by pioneers. If ‘regenerating logged forest’, used later, is correct, then this should be used throughout, including in the Abstract. Assuming this designation IS correct, it then seems unlikely to me that the canopy trees that dominate the results are all or even mostly the ‘early successional species’ whose anatomy and physiology favors drought intolerance. This should be checked in the permanent plots, but I think the implication that the regenerating forest canopy is dominated by pioneers is wrong.

Many thanks for your comments - we agree that it is important to be more careful with these terms throughout the manuscript. Our analysis covers a gradient ranging from lightly to heavily logged forests, as well as canopy growth of fragmented forests affected by oil palm plantations. We are now using the term ‘regenerating logged forests’ throughout the manuscript and have been careful to clarify the source of disturbance being studied.

Although our study focuses on the recovery of logged forests (not ones that are regenerating from cultivation), heavily logged forests are dominated by pioneer species across our study area (Riutta *et al.*, 2018) with acquisitive traits (Both *et al.*, 2019). Specifically, based on inventory data from the 1-ha Global Ecosystem Monitoring (GEM) plots distributed across the logging gradient at SAFE, we found that heavily logged forests had 57.2% of the basal area represented

by pioneer tree species, moderately logged forests 21.5%, and lightly logged forests had 6.9%. We now highlight this information in the manuscript to make it clear that the heavily logged forests are largely dominated by pioneer species. It is correct, however, that the recovering logged forests of this study do retain old-growth forest species, even if their abundance is low. We have made sure to also highlight this aspect in the revised paper.

Secondly, the Abstract implies that reductions in canopy height are due to leaf shedding and (whole tree?) mortality. Both should be obvious from the ground, while branch loss, mentioned later, is less clear from ground level. If you have evidence for branch loss, this should be mentioned in the Abstract and elsewhere, since it appears to an overlooked and neglected impact of droughts.

Agreed, this is an important point which we should have been clearer about in our previous version of the manuscript. Our Leaf Area Index (LAI) estimates are actually Plant Area Index (PAI) estimates from canopy openness measurements. PAI is the sum of LAI and the woody components of the trunk and canopy, including branches. In the revised manuscript we clarify this aspect and have brought in additional field data from the 1-ha GEM plots to show how leaf litter and branch fall changed through time during the El Niño (see revised Fig. 2 in the main text, reproduced below). These data indeed suggest that changes in PAI are likely resulting from a combination of leaf shedding and branch loss (as well as whole-tree mortality). These results have been integrated into the revised manuscript, which now includes a more comprehensive discussion of the impacts of logging and drought on the woody component of the canopy (including branch loss).

Fig. 2: The 2015-16 ENSO effects on tree and canopy field measurements. Time-series of (a) mean field-estimated top-of-canopy height (TCH) in meters, (b) Plant Area Index ($\text{m}^2 \text{m}^{-2}$), (c) mean branch fall ($\text{MgC ha}^{-1} \text{year}^{-1}$) and (d) mean leaf fall ($\text{MgC ha}^{-1} \text{year}^{-1}$) between 2013 and 2018 with the period between both LiDAR surveys (November 2014 - April 2016) which were coincident with the ENSO event highlighted in light grey. Error bars represent relative 95% confidence intervals.

Thirdly, we are told that ENSO events are becoming more severe, but the evidence for this is still fairly weak. You need to give a balanced view on this, with support from the recent literature and CMIP6 projections.

Many thanks for this very important point - we agree that it's important to provide an accurate and balanced summary of how the frequency and intensity of ENSO events has changed in recent years. Recent studies suggest that the frequency (although not the severity) of El Niño is increasing (Cai *et al.*, 2018; Vogel, Hauser and Seneviratne, 2020), including in Southeast Asia (Thirumalai *et al.*, 2017). We have updated the manuscript with these references when we

mention that ENSO events are becoming more frequent, as these studies have used CMIP5 and CMIP6 climate models to explore how global warming is affecting ENSO.

Finally, I found the mixing of review and new results in the Discussion and Conclusions is often confusing. This section should primarily discuss YOUR results, and compare them with the literature where appropriate, but readers need to know what was learned from this study. A synthesis of all the evidence would also be useful, but it should come after the results of this study have been discussed.

Thank you for your comment. We agree that in certain sections of the Discussion it wasn't always clear whether we were referring to our results or those of other studies. We have worked hard to rectify this in the revised paper, which we now feel provides a much clearer overview of our results and how they fit into the broader context of the literature.

REFERENCES

Both, S. *et al.* (2019) 'Logging and soil nutrients independently explain plant trait expression in tropical forests', *The New phytologist*, 221(4), pp. 1853–1865.

Cai, W. *et al.* (2018) 'Increased variability of eastern Pacific El Niño under greenhouse warming', *Nature*, 564(7735), pp. 201–206.

Riutta, T. *et al.* (2018) 'Logging disturbance shifts net primary productivity and its allocation in Bornean tropical forests', *Global change biology*, 24(7), pp. 2913–2928.

Thirumalai, K. *et al.* (2017) 'Extreme temperatures in Southeast Asia caused by El Niño and worsened by global warming', *Nature communications*, 8, p. 15531.

Vogel, M. M., Hauser, M. and Seneviratne, S. I. (2020) 'Projected changes in hot, dry and wet extreme events' clusters in CMIP6 multi-model ensemble', *Environmental research letters: ERL [Web site]*, 15(9), p. 094021.

Reviewer #2 (Remarks to the Author):

Review on manuscript “Recovery of logged forest fragments in a human-modified tropical landscape”

General comments:

The authors present data on forest regeneration after the 2015-16 ENSO event based on canopy height growth across a converted tropical landscapes in Malaysian Borneo using repeat airborne LiDAR surveys, field-based forest inventories and LAI measurements. Their main findings reveal that forest fragmentation and topographic position influence canopy growth, with hilltop forests and small forest patches affected more strongly and pervasively. The study further identified strong edge effects extending over 500 metres into early-successional forest having implications for forest management.

I am convinced that this study is a significant contribution to a better understanding of how climatic events such as reoccurring droughts will influence tropical landscapes in the future aiding guidelines for restoration and is thus of interest to the scientific community and the broader public. While some of the findings discussed in this paper are not completely novel and have been observed before based on field-based inventories, the spatial scope and detail on the interface between remote sensing and forest monitoring are not present for Southeast Asia and rarely across the tropics to my knowledge.

Thank you for your kind words, we are pleased you enjoyed our paper. We agree that its novelty lies largely in the integration of field and remote sensing data to understand how human-modified tropical forests are responding to global change at scale.

Overall, I find the study to be well written and comprehensible. To my understanding the methods and statistical analyses are valid and described in sufficient detail. My only principle critique regards the terminology of riparian forest in this study, which did not

become sufficiently clear from my point of view. Assumably they are being opposed to hilltop forests and roughly follow streamlines, but not every non-hilltop forest is a riparian forest in many tropical landscapes. It may be imaginable in a very rugged terrain, but then the authors should include more information on the relief and altitudinal amplitudes in the study area. Overlaying the topographic position index with the stream network may not be a sufficient determinant as the soil water regime and consecutive effects on water deficit in trees is likely to be quite different in the areas labelled as riparian forests.

The reviewer makes an important point which we agree with and which we have worked hard to clarify in revising our manuscript. The SAFE Project landscape has a varied topography, with the lowlands (100–350 m a.s.l.) largely converted to oil palm plantations and the remaining forest predominantly covering hills rising to almost 1,000 m a.s.l. Most streams within oil palm plantations in the SAFE landscape have a forest buffer zone, the width of which ranges from 12 m to 100 m, although some have no buffer (Gray *et al.*, 2014).

Based on a stream network of the SAFE Project, we show that low topographic position index (TPI) values are generally associated with rivers (Supplementary Fig. 6, 7a). Negative TPI values are likely to be within ~ 75 m from rivers and 50% of the forests with $TPI < -5$ are within 15 m from rivers; thereby the low-lying forests of our study generally represent riparian forests. To further understand how topographic position mediates water availability for trees, we used the DTM to calculate the topographic wetness index (TWI) for each 30 m grid cell using the 'dynatopmodel' R package (Metcalf, Beven and Freer, 2015). High TWI values indicate grid cells with topographic characteristics favourable for accumulating higher levels of soil moisture, and vice versa (Muscarella *et al.*, 2020). When we compare TPI and TWI values across SAFE, we clearly see that areas with more convex curvature and steeper slopes (higher TPI) tend to have lower TWI values, and vice versa (Supplementary Fig. 7b; Spearman correlation = -0.81, P-value < 0.0001). Moreover, recent work at SAFE has shown that variation in TPI across this landscape is strongly related to variation in microclimate and nutrients (Jucker *et al.*, 2018; Swinfield *et al.*, 2020). So while there will be some residual variation in water availability among different forest patches characterised by low TPI values, in most cases this will be swamped by the much larger variation between forests in valleys (low TPI) and those on ridge-

tops (high TPI), which is the focus of our analysis. We have clarified this in the revised manuscript.

Supplementary Fig. 6: River network across the SAFE landscape. A portion of the Topographic Position Index (TPI) map across the SAFE landscape. The black pixels represent TPI values < -5 . The grey, blue and orange pixels are TPI > -5 , with the orange pixels representing the largest TPI values (hilltops). White pixels are NA data due to the presence of clouds. The juxtaposed red dashed lines correspond to rivers from a stream network data for the SAFE Project. The figure demonstrates that areas closer to rivers tend to be areas with low TPI and most of the low-lying areas represent riparian forests.

Supplementary Fig. 7: Relationship of TPI with distance from rivers and water availability. Topographic Position Index (TPI) variation with (a) distance from rivers in meters and (b) topographic wetness index (TWI). The horizontal dashed line represents a TPI value of -5, and the red solid line represents a smoothing loess function indicating that low-lying areas - negative low TPIs - are more likely to be within ~ 75 m from rivers.

The conclusion and last paragraphs of the discussion are very insightful, but perhaps too detailed, slightly repetitive and at times a bit far-stretched particularly regarding the importance of riparian buffer zones given that soil water deficits have not been measured and the classification of riparian forest remains not concrete. Parts of the conclusion appear more like an introduction to physiological mechanisms behind the observed patterns or explanations that seems better placed at the beginning of the manuscript.

Thank you for your comment. We agree that parts of the conclusions read a little bit like an introduction, so in revising our paper we have restructured and rewritten the conclusions to better summarise the key take home messages of our study.

Regarding the importance of riparian forests and buffer zones, as discussed above, our additional analyses support the conclusions that low TPI values are generally associated with wetter areas of the landscape. We acknowledge that there will be some residual variation in water availability among different forest patches characterised by low TPI values, but in most cases this will be

swamped by the much larger variation between forests in valleys (low TPI) and those on ridge-tops (high TPI), which is the focus of our analysis. In revising our paper, we have scaled back the focus on the potential benefits of retaining and restoring riparian forests, some of which we agree went beyond the scope of our study. However, we still feel that our results have important implications for understanding the impacts of climate change on human-modified tropical forests and for designing more resilient and effective riparian buffer zones. We have therefore retained some of the recommendations we make in conclusions, such as the importance of maintaining wide riparian buffers in logged and converted forest landscapes.

Specific comments:

Unfortunately, there are no line and page numbers in the pdf of the manuscript so I will copy the passage I'm referring to here and upload a file where the passages are marked. I hope it will be comprehensible this way.

Apologies for this. We have made sure that the revised manuscript has line and page numbers.

Abstract

- Perhaps the study title could include a brief mention on the type of recovery investigated i.e. after ENSO event/ post-drought or similar.

Thank you for your comment. We agree and have updated the title to "Recovery of logged forest fragments in a human-modified tropical landscape during the 2015-16 El Niño".

- First sentence: I find using the term "loss of nature" is a bit unassertive and generalized as actually habitat transformation is taking place where it is possible to argue nature still exists. Maybe using natural habitat, pristine environment or primary forest would make it clearer.

Agreed. We have changed this to 'primary forests'.

- Sentence 6: Canopy height decrease seems not a direct logical cause of reduced growth. Maybe the authors can add stagnation or rephrase the sentence slightly.

Agreed, we should have made this clearer. We have rephrased the sentence and used “loss in canopy height” throughout the manuscript - given that changes in canopy height result from a combination of tree growth and mortality, as well as changes in Plant Area Index (PAI) resulting from leaf shedding and branch loss.

- Sentence 8: Should it read: will be slow to recover even if protected”? Or will they recover faster if not protected?

It should read: small patches of protected forests will be slower to recover due to effects of fragmentation on the canopy. We have clarified this in the revised manuscript.

Introduction

- Line3: CO₂ in subscript

Corrected, thank you.

Material and methods

Mapping topography and distances from edge

- In the investigated study area there were no other land-use types than oil palm plantations, open clear-cut areas and forests present? Such as tree plantations or other agricultural use?

Our initial and first approach was to classify land-use types within the SAFE landscape - which helped us confirm our previous visual analysis of the high-resolution Pleiades image we used to identify the oil palm boundaries. Although there are some other plantations within the SAFE landscape (e.g. acacia), these are small in extent and are embedded within the vast oil palm plantations. There are also some villages and isolated buildings within these oil palm plantations, but these areas were all removed from our analysis. We attempted to minimise potential confounding factors, thus we removed all other pixels except regenerating natural forests from the analyses.

- The definition of riparian forests remained unclear to me. Did these areas have a certain distance to rivers or other water bodies? Where they periodically flooded? How much percentage of the forests with TPI < -5 is in close proximity to a river or stream? In these areas likely not only water supply but also soil nutrient supply may vary.

This is an important point we have worked hard to clarify in the revised manuscript (see response to a similar question raised above). The SAFE Project landscape has a varied topography with the lowlands (100–350 m a.s.l.) almost entirely converted to oil palm plantations and the remaining forest predominantly covering hills rising to over 1,000 m a.s.l. Most streams within oil palm plantations in the SAFE landscape have a forest buffer zone, the width of which ranges from 12 m to 100 m, although some have no buffer (Gray *et al.*, 2014).

Based on a stream network for the SAFE Project, we demonstrate that low topographic position index (TPI) values are often associated with rivers (Supplementary Fig. 6; 7a). Negative TPI values are likely to be within ~ 75 m from rivers and 50% of the forests with TPI < -5 are within 15 m from rivers; thereby the low-lying forests of our study generally represent riparian forests. To better understand how topographical conditions mediate water availability for plants, we used the DTM to calculate the topographic wetness index (TWI) for each 30 m grid cell using the 'dynatopmodel' R package (Metcalf, Beven and Freer, 2015). High TWI values indicate grid cells with topographical characteristics favourable for accumulating higher levels of soil moisture, and vice versa (Muscarella *et al.*, 2020). Areas with more convex curvature and steeper slopes (higher TPI) tend to have lower values for the TWI (Supplementary Fig. 7b; Spearman

correlation = -0.81, P-value < 0.0001). Moreover, recent work at SAFE has shown that variation in TPI across this same landscape is strongly related to variation in microclimate and nutrients (Jucker *et al.*, 2018; Swinfield *et al.*, 2020). So while there will be some residual variation in water availability among different forest patches characterised by low TPI values, in most cases this will be swamped by the much larger variation between forests in valleys (low TPI) and ridge-tops (high TPI) which is the focus of our analysis. We have clarified this in the revised manuscript.

Field data for validation and mechanism elucidation

**- For how many trees within the permanent the height and crown extent was measured?
Was a rangefinder or Vertex used for the height measurements?**

Tree height (H) and crown area (CA) were measured for a total of 3248 trees in the 1-ha Global Ecosystems Monitoring (GEM) permanent plots. We used these data to fit allometric equations for predicting the H and CA of trees that were not measured ($n = 10393$; Supplementary Fig. 8). H was measured using an Impulse 200 laser rangefinder (Laser Technology Inc., Colorado, USA). CA was estimated from ground-based mapping of the horizontal crown projection of each tree, using the laser Field-Map technology (IFER, Ltd., Jílové u Prahy, Czech Republic; for details of the technology, see (H'edl *et al.*, 2009).

- “The field-estimated canopy height agreed well with the LiDAR-based canopy height for both LiDAR surveys.” This could be specified a bit giving a measure of variance or model fit.

Agreed. We have added 95% confidence intervals to the fitted plots shown in Supplementary Fig. 9. We have also calculated and reported the Residual Standard Error (RSE) as a measure of model fit.

- "...LAI change were estimated as the difference between the end (February 2017) and the beginning (November 2014)." There is no expected seasonal influence of rainy or drier season on LAI in the study area? Are there deciduous or brevideciduous species present?

The climate in the region is aseasonal. Approximately 12% of months experience rainfall <100 mm / month, but there are no distinct, regularly occurring dry and wet seasons (Newbery *et al.*, 1999). We have updated Fig. 2b of the manuscript with additional pre El Niño PAI data, which shows that although there is some season variation in PAI this is much smaller than the loss in PAI that occurred following the 2015-2016 El Niño event. We have also included data on branch fall and leaf litter fall from the GEM plots which shows how both branch and leaf losses contributed to the observed declines in PAI (Fig. 2c, d).

Fig. 2: The 2015-16 ENSO effects on tree and canopy field measurements. Time-series of (a) mean field-estimated top-of-canopy height (TCH) in meters, (b) Plant Area Index ($m^2 m^{-2}$), (c) mean branch fall ($MgC ha^{-1} year^{-1}$) and (d) mean leaf fall ($MgC ha^{-1} year^{-1}$) between 2013 and 2018 with the period between both LiDAR surveys (November 2014 - April 2016) which were

coincident with the ENSO event highlighted in light grey. Error bars represent relative 95% confidence intervals.

- How often were the hemispherical photos taken for the continuous measurements in the GEM plots?

The collection was repeated approximately every three months. We have clarified this in the manuscript.

- Figure 1 Header: Perhaps better to use “climatic variables” instead of climate

Agreed, thank you.

Discussion

- Line 4: “...when temperatures exceeded 2.1 °C and VPD exceeded 140% of the local long-term average.” This is not entirely clear to me. Which timeframe was counted? Did the maximal or average temperature/VPD exceed the long-term average by 140% or was it 140% of this average?

Apologies for the confusion. What we meant is that the highest temperatures and VPD during the El Niño (March 2016) exceeded the long-term average of non-El Niño years (2013-2014). We have been more specific in the manuscript as follows: “...with the highest temperatures and VPD during the El Niño (March 2016) exceeding the long-term average of non-El Niño years (2013-2014) by 2.1°C and 140%, respectively (Fig. 1b, c).”

- “Fragmentation effects increasing exponentially with proximity to oil palm plantations are consistent...”

Corrected. Thank you.

Forest recovery in human modified tropical landscapes

- Perhaps rephrase to: This result supports the high growth rates observed in secondary forests in the Neotropics

Thanks for the suggestion. We have adopted it in the revised manuscript.

- This statement as stand-alone should be used more cautiously as the study actually identified hydraulic traits to influence the probability of mortality among species and was conducted on a subset of 20-50 species from seasonally dry tropical forest. There is plenty of robust evidence from tropical rainforests that tree size and other plant functional traits are linked with vulnerability to drought and mortality. E.g. see Anderegg 2016, 2019, Aleixo 2019.

This is true and thanks for your comment. Our emphasis was to demonstrate the importance of landscape information for predicting the demographic impacts of drought-related mortality. We have modified this section of the manuscript as follows: “...The high abundance of pioneer species - which in heavily logged forests at SAFE make up > 50% of the total forest basal area - can also make regenerating forests more vulnerable to higher temperatures and drought, as pioneer species tend to be less well suited to coping with the high evaporative demands and lower soil water availability that characterise ENSO events (Ouédraogo *et al.*, 2013; Uriarte *et al.*, 2016). However, our results suggest that far away from forest edges, regenerating logged forests continued to grow in height during the 2015-16 ENSO.”.

Conclusion

- “Droughts can lead to immediate tree death because of hydraulic failure, or slower tree death because of carbon starvation or a combination of both mechanisms, and can be synergistically coupled to the effects of storms, windthrows, pathogens and insects” That seems to belong rather in the introduction.

Agreed. We have moved this sentence from the conclusions to the main body of the discussion (see line 271 in the revised manuscript).

Figures

- Figure 5: It would be good to include a legend for the hilltop and valley colour code. The figure already has a lot of white space which could perhaps be improved.

Agreed. We have updated the figure. Thank you.

Fig. 5: Fragmentation and topographic effects on canopy change during the 2015-16 ENSO event.

Supplementary Information:

- Figure S2: The axis label size should be increased

Thank you. We have updated the figure.

Supplementary Fig. 5: Spatial distributions of environmental variables. Spatial distribution of (a) Pre-ENSO Top-of-canopy height (TCH), (b) Topographic Position Index (TPI) and (c) distance of forests from oil palm plantations covered by repeated airborne LiDAR data across the SAFE landscape in Malaysian Borneo.

- Figure S5: The field estimated TCH measurements were conducted nearly one year after the LiDAR ones when the peak of the ENSO-event was occurring. Do the authors expect any influence of the

We first tested the correlation between the field data collected in December 2015 and Feb 2017 with the LiDAR TCH measurements and found a much closer relationship when using the field data from 2017. Despite the possible influences of forest recovery post-ENSO on this relationship, we tried to minimise these influences by choosing the field data with the closest relationship to our LiDAR observations.

- Figure S6: I did not find the figure mentioned in the text. It's a pity as I find the results very exciting.

Thank you for pointing it out. Supplementary Fig. 10 (former Figure S6) is now mentioned in the manuscript as follows “A predicted Δ map depicts the canopy change variation across the entire studied landscape with varying TPI and distance from oil palm plantations (Supplementary Fig. 10)”.

REFERENCES

Gray, C. L. *et al.* (2014) ‘Do riparian reserves support dung beetle biodiversity and ecosystem services in oil palm-dominated tropical landscapes?’, *Ecology and evolution*, 4(7), pp. 1049–1060.

Hedl, R. *et al.* (2009) ‘A new technique for inventory of permanent plots in tropical forests: a case study from lowland dipterocarp forest in Kuala Belalong, Brunei Darussalam’, *Blumea - Biodiversity, Evolution and Biogeography of Plants*, 54(1-2), pp. 124–130.

Jucker, T. *et al.* (2018) ‘Canopy structure and topography jointly constrain the microclimate of human-modified tropical landscapes’, *Global change biology*, 24(11), pp. 5243–5258.

Metcalfe, P., Beven, K. and Freer, J. (2015) ‘Dynamic TOPMODEL: A new implementation in R and its sensitivity to time and space steps’, *Environmental Modelling & Software*, 72, pp. 155–172.

Muscarella, R. *et al.* (2020) ‘Effects of topography on tropical forest structure depend on climate context’, *The Journal of ecology*. Edited by T. Jucker, 108(1), pp. 145–159.

Newbery, D. M. *et al.* (1999) ‘The ecoclimatology of Danum, Sabah, in the context of the world’s rainforest regions, with particular reference to dry periods and their impact’, *Philosophical transactions of the Royal Society of London. Series B, Biological sciences*, 354(1391), pp. 1869–1883.

Ouédraogo, D.-Y. *et al.* (2013) ‘Slow-growing species cope best with drought: evidence from long-term measurements in a tropical semi-deciduous moist forest of Central Africa’, *The Journal of ecology*. Edited by M. Turnbull, 101(6), pp. 1459–1470.

Powers, J. S. *et al.* (2020) 'A catastrophic tropical drought kills hydraulically vulnerable tree species', *Global change biology*, 26(5), pp. 3122–3133.

Swinfield, T. *et al.* (2020) 'Imaging spectroscopy reveals the effects of topography and logging on the leaf chemistry of tropical forest canopy trees', *Global change biology*, 26(2), pp. 989–1002.

Uriarte, M. *et al.* (2016) 'Impacts of climate variability on tree demography in second growth tropical forests: the importance of regional context for predicting successional trajectories', *Biotropica*, 48(6), pp. 780–797.

Reviewer #3 (Remarks to the Author):

I was asked to review the manuscript written by Nunes et al. This manuscript demonstrates with repeat lidar that forest recovery during ENSO varies spatially and gives new insights into fragmentation of regenerating forests. Whereas the methods themselves are mostly building on existing work and methods, the context and conclusions are important. I was happy to see that the authors also included some recommendations regarding forest (fragmentation) management. Overall, this was a well written manuscript, but I found some sections overly detailed, whereas other sections did not have enough (or confusing) details to fully understand the methods that were used, which is critical. I did enjoy the discussion and I thought that was excellent. I invite the authors to respond to my concerns.

We really appreciate the reviewer's encouraging words and valuable comments which we feel have considerably helped us improve our paper. Below we provide detailed responses to the referee's main concerns and indicate how these have been addressed in the revised manuscript.

My main concerns are related to the lidar analysis and the detail in the manuscript has not convinced me that these 2 datasets are interoperable.

1) What is the beam divergence (and exit diameter) of both sensors? The authors report the footprint (I assume at top of canopy or ground) of 40cm vs 180 cm. They then use lastools to generate a DTM with 2m resolution. However, if the beam divergence is significantly different (I found a beam div of 0.22 mrad for NERC and 0.5 mrad for GAO), then the sensor with the smaller beam divergence will be able to penetrate the canopy more easily, having a higher probability of actually detecting a ground return. The other way around, the sensor with the larger beam divergence might not always detect the actual ground, but

some averaged values of elements near the ground. This is quite important as this is the baseline for tree height.

2) Why is the NERC reported point density 4pts/m²? In Jucker et al 2018 (<https://onlinelibrary.wiley.com/doi/epdf/10.1111/gcb.14415>), this dataset has 15.3pts/m²? Jucker et al 2018 mentions 4 pts per laser pulse. This links back to my previous comment on data-operability as the NERC data might now have 4 times more points and smaller beam divergence (to be confirmed by the authors). The authors also mention that "only data with point density above 10pts/m² were used"; but the reported point density for the 2016 flight is only 2pts/m²?

Thank you for your comments and apologies for the lack of details regarding the LiDAR datasets. Below we address comments 1 and 2 together. We have updated the manuscript and Supplementary Information with the following details.

The interoperability of the two LiDAR datasets was indeed something we were initially concerned with and put a lot of effort into mitigating in our analysis. As the reviewer points out, the beam divergence of the two sensors is different (the numbers reported by the reviewer are correct). Similarly, the mean point density of the 2014 NERC data was 13.2 points m⁻² (± 13.2 points m⁻² standard deviation) whereas the 2016 GAO's mean point density (2016) was 4.1 points m⁻² (± 2.2 points m⁻² standard deviation) (as we now clarify in the revised manuscript). However, despite these differences in beam divergence and point density, based on extensive sensitivity analyses we have conducted as part of this review process, we believe that the two datasets can be used to robustly estimate canopy height changes across the SAFE landscape and test how these are affected by forest fragmentation and topography. Here we start by discussing why we do not feel that differences in beam divergence between the two sensors would have had a big impact on our results and then provide evidence of how our results are robust to differences in point density.

The assessment of the potential effect of beam divergence on tree height estimation can undermine the importance of several factors involved in ground point detection. The GAO uses higher-wattage lasers with a larger beam divergence (0.5 mrad), which combine to increase the chance of ground detection in thick tropical understories. To minimise errors in the fusion of

both LiDAR datasets, we (i) created a common digital terrain model (DTM) using a combination of ground returns from both LiDAR surveys, (ii) coarsened pixel resolution to reduce uncertainties due to artefacts of repeat LiDAR data such as wind direction and within-canopy variation, (iii) restricted our analysis to areas with high point density in the 2014 NERC LiDAR survey and (iv) tested the sensitivity of our results to spatial variation in point density in the 2016 GAO LiDAR survey.

Firstly, a common digital terrain model (DTM) at 1 m resolution was created using a combination of ground returns from both surveys. Using a TIN-densifying algorithm available in “lasground” tool, LiDAR returns were classified as ground and non-ground, and their heights above ground were calculated by subtracting the elevation of the resulting DTM underneath each of them. Then a canopy height model (CHM) representing the height of vegetation was generated separately for each survey, following the methodology outlined by Khosravipour et al. (2014), and top-of-canopy height aboveground (TCH) was initially calculated as the mean CHM at 2 m resolution. It has been demonstrated that TCH, as measured by LiDAR, is a useful metric for estimating structural attributes of natural tropical forests and is relatively insensitive to sensor and flight specifications (Asner and Mascaro, 2014).

Variance of top-of-canopy height (TCH) change can be overestimated owing to artefacts of repeat LiDAR data such as wind direction and within-canopy variation. We used a subset of TCH change data to investigate how bilinear resampling would affect the mean and variance of TCH change. Mean values were not affected by coarsening pixels from 2 m to lower resolutions, however standard deviation (SD) was significantly reduced; SD of 2-m resolution TCH change was 4.9 m, with a significant reduction to 2.1 m for 36-m resolution pixels - when SD was no longer affected by pixel size (Supplementary Fig. 1a, b). The final TCH map was then coarsened from 2 m to 30 m resolution following (Asner *et al.*, 2018)) given the similar SD values of TCH change to the lower resolution data.

Supplementary Fig. S1: Reducing uncertainties in top-of-canopy height estimates. Changes in top-of-canopy height (TCH) can be overestimated owing to artefacts of repeat LiDAR data such as wind direction and within-canopy variation. From a subset of the repeat LiDAR data, **(a)** shows a reduction in standard deviation (SD) and **(b)** the consistent mean values of TCH change with resampling to lower resolution pixels.

LiDAR estimates of TCH can be affected by low point density (Gobakken and Næsset 2008). To test for potential biases arising from differences in point density, we first tested how point density of the 2014 NERC LiDAR survey affected the LiDAR-based TCH change estimates (Supplementary Fig. 2). In comparison with the field-based TCH changes of 0.23 m (95% CI: -0.30 – 0.77 m) (shaded area, Supplementary Fig. 2), an underestimate of tree height associated with point density < 10 points m^{-2} in the NERC dataset may have contributed to an overestimation of TCH change. Thus, we removed these pixels ($\sim 70\%$ of the dataset) and only data with point density ≥ 10 points m^{-2} in the NERC dataset were used to avoid canopy height change overestimation.

Supplementary Fig. 2: NERC’s point density effects on top-of-canopy height (TCH) change estimates. Point density of the 2014 NERC LiDAR survey versus top-of-canopy height (TCH) change in meters between the two LiDAR surveys. Each point represents a 30 m resolution pixel. The red line depicts a smoothing loess function and the shaded area represents the field-based TCH change of (95% CI: -0.30 – 0.77 m).

We also assessed the influence of point density in the GAO data on TCH change estimation by repeating the same analysis using GAO’s low point density (< 2 points m^{-2}) and GAO’s high point density (> 2 points m^{-2}) datasets. We demonstrate below that (i) although there are some differences in the spatial distribution of areas with high and low GAO’s point density (Supplementary Fig. 3a, b,c), there is also enough overlap to test for point density effects on tree height change estimations, and (ii) that even if we remove areas with low point density the results are unchanged (Supplementary Fig. 4 versus Fig. 5 of the manuscript), most probably because of the wider beam divergence and higher-wattage lasers. Although low point density areas in the GAO data predominantly cover short forests and high point density data cover tall forests (Supplementary Fig. 3a), our canopy height growth predictions are similar using only high point density data, which is unlikely to happen if point density were affecting these patterns. We did observe less-pronounced edge effects when using the high point density data only

(Supplementary Fig. 4 versus Fig. 5 of the manuscript), however this is due to the fact that most forests neighbouring oil palm plantations were covered by low point density data (Supplementary Fig. 3c) and thus we have an insufficient number of pixels to robustly fit the prediction models.

Supplementary Fig. 3: Spatial distribution of GAO's point density. distribution analysis of areas with low point density (<2 points m^{-2} ; red) and high point density (>2 points m^{-2} ; blue) in the GAO data in relation to (a) top-of-canopy height (TCH) in meters, (b) topographic position index (TPI) and (c) distance from oil palm plantations in meters.

Supplementary Fig. 4: Canopy height change predictions using GAO’s high point density dataset. Predicted effects of distance from oil palm plantations and topographic position index (TPI) on top-of-canopy height (TCH) growth per year in forests of different successional stages during the 2015-2016 El Niño in Malaysian Borneo. These predictions were based on a subset of the GAO’s point density above 2 points m^{-2} to investigate whether the predictions were affected by point density in the GAO data. Median LiDAR-based TCH change per year was predicted by nonlinear modelling and its 95% confidence interval (shaded area). Short, medium and tall canopies were 5 m, 20 m and 35 m in top-of-canopy height (TCH) measured in the first LiDAR survey (November 2014). Valley-bottom (blue) and hilltop (orange) curves correspond with TPI values of -8.2 and 9.0, respectively, from the 5th and 95th quantiles.

3) Figure S1A: Is the caption of that figure correct? I think the authors should include these uncertainties in the analysis and figures and introduce error bars when reporting lidar TCH change.

The caption of Supplementary Fig. 1 was not correct - apologies for the mistake and thank you for the comment. The aim of the figure was to show that by coarsening our data from 2 m to 30 m following (Asner *et al.*, 2018), we found a significant reduction in uncertainties in TCH change variance due to artefacts of repeat LiDAR data such as wind direction and within-canopy

variation with no significant effects on the mean of TCH change values. We corrected it in the Supplementary Information (Supplementary Fig. 1a, b).

We have also introduced error bars and updated the manuscript when reporting LiDAR TCH change. Many thanks for your suggestion.

I am also a bit concerned about the "validation", or at least the use of this term. Whereas lidar gives you more or less a direct estimate of height; how can an estimate through an allometry (field height) be serving as a validation? What essentially is measured in the field is mostly DBH, not height. At the very least, the uncertainty of these allometry height values should be taken into account. How are these field H measurements done? (There is a sentence "canopy height growth: canopy height was estimated from the forest plot dataset .."). How was this done/measured and how does this link to the H-DBH allometry mentioned in the previous paragraph?

Tree height (H) and crown area (CA) for individuals with no height and crown measurements in the field were estimated using non-linear least-square allometric models locally calibrated from 3248 tree H - DBH and CA - DBH relationships within the Global Ecosystems Monitoring (GEM) permanent plots across the same landscape (Eq. S1 and S2; Supplementary Fig. 8). We used these data to fit allometric equations for predicting the H and CA of trees that were not measured ($n = 10393$). H was measured using an Impulse 200 laser rangefinder (Laser Technology Inc., Colorado, USA). CA was estimated from ground-based mapping of the horizontal crown projection of each tree, using the laser Field-Map technology (IFER, Ltd., Jílové u Prahy, Czech Republic; for details of the technology, see Hédli et al. 2009). This was included in the Material and methods to clarify how the measurements and estimations of tree height and crown area were done.

Regarding the use of the term ‘validation’, we agree that the field data we have collected can only really be used to corroborate and interpret the results obtained from the repeat LiDAR flight - not to robustly validate them. To avoid this confusion, we have therefore renamed this section of the manuscript to “Using field data to elucidate the mechanisms driving changes in canopy height” and have removed any reference to the use of field data for validation from the Results

section. That being said, while the field data may only be of limited use for directly validating the LiDAR results, we still feel they are hugely valuable for exploring the range of mechanisms driving canopy height changes during the ENSO event. In the revised manuscript we have built on this aspect of the paper by bringing in additional field data on temporal trends in branch fall and leaf litter fall rates from the 1-ha GEM plots (Fig. 2).

Lastly, related to the reviewer's comments, in the revised discussion we now also specify that “...we were unable to explain a large amount (~57%) of the variation in canopy growth. This residual variance could be associated with varying species composition of the plots (as sensitivity to drought can vary considerably among species), as well as canopy variables that we were unable to quantify or adequately account for. These include the uncertainties in height and crown area predictions used to estimate field canopy height, as well as PAI estimations based on canopy openness measurements. Additionally, LiDAR-based canopy heights were obtained using different sensor configurations and flight parameters, which can affect canopy height estimation (Roussel *et al.*, 2017). To minimise this effect, we restricted our analysis to areas with high point density in the 2014 LiDAR survey. We also tested the sensitivity of our results to spatial variation in point density in the 2016 LiDAR survey, but found no evidence that this affected our conclusions”.

I have a similar issue with the term LAI for validation. It appears that what the authors did measure was PAI. However, this makes it more interesting as the effect of forest recovery on the woody and leaf component of PAI might not be equal. The last sentence in the 2nd paragraph of "correlations between ... LAI change" suggest that the sharp drop in LAI in 2016 is due to high temperatures, but looking at fig 2A and B this sharp decline might already happen earlier, but the loss of LAI/PAI during El Nino might have been partly compensated by tree growth (and therefore increase in the woody component).

Many thanks for this extremely valuable comment. Most of the optical instruments have only one visible band and cannot distinguish leaf and woody components; thus, the results obtained are indeed plant area index (PAI), which is the sum of the leaf and Woody Area Index (WAI). We are now using PAI as a function of canopy openness based on the method of (Kalacska *et al.*,

2005) developed for “Palo Verde National Park” during the wet season in Costa Rica. PAI change has now a much closer relationship to TCH change than our previous LAI change estimates. PAI change + field TCH change together account for 43% of the total LiDAR THC change variation (Fig. 3).

Fig. 3: Correlation between LiDAR-derived and field-estimated change in canopy structure during the 2016-15 ENSO event. LiDAR-based top of canopy height (TCH) change (m) versus (a) field-estimated TCH change (m) and (b) Plant Area Index (PAI) change ($\text{m}^2 \text{m}^{-2}$) between November 2014 and February 2017. Each dot represents one permanent plot and the red lines represent a multiple linear regression, $\text{LiDAR-based TCH change} \sim \text{PAI change} + \text{field-estimated TCH change}$, with the shaded grey area depicting the 95% CI.

We also included the branch fall and leaf litter time-series to demonstrate that branch and leaf losses contributed to the decline in PAI during the El Niño (Fig. 2c, d). The PAI decrease during the El Niño was coincident with an increase in branch fall early 2016 (F-value = 3.14, P-value = 0.0337) and in leaf litter throughout 2015 (F-value = 44.96, P-value < 0.0001).

Fig. 2: The 2015-16 ENSO effects on tree and canopy field measurements. Time-series of (a) mean field-estimated top-of-canopy height (TCH) in meters, (b) Plant Area Index ($\text{m}^2 \text{m}^{-2}$), (c) mean branch fall ($\text{MgC ha}^{-1} \text{year}^{-1}$) and (d) mean leaf fall ($\text{MgC ha}^{-1} \text{year}^{-1}$) between 2013 and 2018 with the period between both LiDAR surveys (November 2014 - April 2016) which were coincident with the ENSO event highlighted in light grey. Error bars represent relative 95% confidence intervals.

Smaller comments:

- Intro, first sentence "could offset much of the ..". It would be good to have some number of % here. How long before we will see an effect of restoration; does growth of very young trees offset much of these carbon emissions?

Thank you for your suggestion. We have been more specific and referred to natural forest regrowth as follows: "Natural forest regrowth could offset 25 % of the current annual fossil fuel emissions (10 GtC year^{-1}) (Le Quéré *et al.*, 2018; Cook-Patton *et al.*, 2020), helping to stabilize atmospheric CO_2 concentrations as we transition to a low-fossil-fuel economy in the coming decades (Houghton, Byers and Nassikas, 2015)."

- Intro, start second paragraph ".. effects may be amplified ..". But you only do your analysis in the ENSO period; what is the relevance of this suggestion?

Thanks for the suggestion and we agree that it is irrelevant to say that the fragmentation effects may be amplified given the nature of our data. We have started the second paragraph with the following evidence: "The effects of fragmentation have been amplified by dry conditions experienced during El Niño Southern Oscillation (ENSO) events (Qie et al., 2017)", as we aim to have a paragraph on the combined effects of fragmentation and climatic events on forests in the following sentences.

- Intro, 3rd paragraph "Differences in forest composition .. tolerance". This very recent discussion in <https://www.nature.com/articles/s41467-020-17214-4> might be worth including here?

That's a great read and reference. Thanks for the suggestion.

- Study site: It would be good to have map of the area (in the supplementary) with an overlay of the extent of the 2 lidar flights

Although we did not explicitly acknowledge it on the paper, SupplementaryFig. 10 depicts the TCH, TPI and predicted TCH change maps of the area. We now refer to the figure in the Material and methods, as well as in the Results section.

- Precipitation: Is there only 1 weather station? How do you then get information of precipitation in hilltops vs low-lying areas. Your 3rd objective seem to indicate that water availability is a factor, so how was this extrapolated over the extent of the study area?

The precipitation data was to demonstrate that a running 30-day precipitation time-series from daily precipitation measurements in the SAFE Project shows decreased precipitation between January 2015 and April 2016 linked to ENSO.

Based on a stream network of the SAFE Project, we show that low topographic position index (TPI) values are generally associated with rivers (Supplementary Fig. 6, 7a). Negative TPI values are likely to be within ~ 75 m from rivers and 50% of the forests with $TPI < -5$ are within 15 m from rivers; thereby the low-lying forests of our study generally represent riparian forests. To further understand how topographic position mediates water availability for trees, we used the DTM to calculate the topographic wetness index (TWI) for each 30 m grid cell using the 'dynatopmodel' R package (Metcalf, Beven and Freer, 2015). High TWI values indicate grid cells with topographic characteristics favourable for accumulating higher levels of soil moisture, and vice versa (Muscarella *et al.*, 2020). When we compare TPI and TWI values across SAFE, we clearly see that areas with more convex curvature and steeper slopes (higher TPI) tend to have lower TWI values, and vice versa (Supplementary Fig. 7b; Spearman correlation = -0.81, P-value < 0.0001). Moreover, recent work at SAFE has shown that variation in TPI across this landscape is strongly related to variation in microclimate and nutrients (Jucker *et al.*, 2018; Swinfield *et al.*, 2020). So while there will be some residual variation in water availability among different forest patches characterised by low TPI values, in most cases this will be swamped by the much larger variation between forests in valleys (low TPI) and those on ridge-tops (high TPI), which is the focus of our analysis. We have clarified this in the revised manuscript.

-'3000 randomly selected pixels.. Memory demanding': How memory demanding? I am surprised this is still an issue when cloud computing and computing clusters are often available. The authors should at least provide a figure (in supplementary) that 3000 pixels is enough.

Thank you for this valuable comment. Our explanation regarding our approach to select subsets of the data was too simplistic. Running a model with spatial autocorrelation structure is also very time demanding. Running the model based on a 3,000-pixel subset can take up to 4 - 5 hours,

whereas a 5,000-pixel subset may take 10 - 11 hours. However, we absolutely agree that it was our mistake to consider that 3,000 pixels were enough without giving sufficient evidence.

We explored this in detail and have clarified this in the revised manuscript.

To test whether a 5,000-pixel subset is sufficient to estimate unbiased parameter values, we first ran 24 randomised permutations of the Equation 1 with the spatial autocorrelation structure for randomly selected 3,000-, 4,000- and 5,000-pixel subsets from the 36,679-pixel dataset. We then generated the mean and coefficient variation (CV %) of parameter values for the assessment of model stability with increasing subset sizes (Table S2). We also ran 24 randomised permutations of the Equation 1 with no spatial autocorrelation structure to investigate whether mean parameter values differed from parameter values when using the full dataset (Table S3). Given (i) the considerably smaller CV for the 5,000-pixel subset and (ii) the similar mean parameter values and confidence intervals of 24 5,000-pixel subsets to the full dataset's parameters, we demonstrate the consistency of 5,000-pixel subsets to predict canopy height change across the landscape.

- Will (or is) the data and code publicly available?

We will make the LiDAR-based canopy height for both LiDAR surveys, topographic position index (TPI) and distance from oil palm plantations data available.

REFERENCES

Asner, G. P. *et al.* (2018) 'Mapped aboveground carbon stocks to advance forest conservation and recovery in Malaysian Borneo', *Biological conservation*, 217, pp. 289–310.

Cook-Patton, S. C. *et al.* (2020) 'Mapping carbon accumulation potential from global natural forest regrowth', *Nature*, 585(7826), pp. 545–550.

Gobakken, T. G. and Næsset, E. N. (2008) 'Assessing effects of laser point density, ground sampling intensity, and field sample plot size on biophysical stand properties derived from

airborne laser scanner data', *Canadian journal of forest research. Journal canadien de la recherche forestiere*. doi: 10.1139/X07-219.

H'edl, R. *et al.* (2009) 'A new technique for inventory of permanent plots in tropical forests: a case study from lowland dipterocarp forest in Kuala Belalong, Brunei Darussalam', *Blumea - Biodiversity, Evolution and Biogeography of Plants*, 54(1-2), pp. 124–130.

Houghton, R. A., Byers, B. and Nassikas, A. A. (2015) 'A role for tropical forests in stabilizing atmospheric CO₂', *Nature climate change*, 5(12), pp. 1022–1023.

Jucker, T. *et al.* (2018) 'Canopy structure and topography jointly constrain the microclimate of human-modified tropical landscapes', *Global change biology*, 24(11), pp. 5243–5258.

Kalacska, M. E. R. *et al.* (2005) 'Effects of Season and Successional Stage on Leaf Area Index and Spectral Vegetation Indices in Three Mesoamerican Tropical Dry Forests¹', *Biotropica*, 37(4), pp. 486–496.

Le Quéré, C. *et al.* (2018) 'Global Carbon Budget 2018', *Earth System Science Data*, 10(4), pp. 2141–2194.

Metcalf, P., Beven, K. and Freer, J. (2015) 'Dynamic TOPMODEL: A new implementation in R and its sensitivity to time and space steps', *Environmental Modelling & Software*, 72, pp. 155–172.

Muscarella, R. *et al.* (2020) 'Effects of topography on tropical forest structure depend on climate context', *The Journal of ecology*. Edited by T. Jucker, 108(1), pp. 145–159.

Swinfield, T. *et al.* (2020) 'Imaging spectroscopy reveals the effects of topography and logging on the leaf chemistry of tropical forest canopy trees', *Global change biology*, 26(2), pp. 989–1002.

Reviewer comments second round -

Reviewer #2 (Remarks to the Author):

Revision of manuscript "Recovery of logged forest fragments in a human-modified tropical landscape during the 2015-16 El Niño"

I would like to thank the authors for this very mindful and thorough revision of the manuscript. I believe that the main concerns raised by the reviewers were answered satisfactorily in much detail and changes well incorporated in the manuscript. Sections of the discussion and conclusion that were lacking some details were supplemented and accordingly improved. Adjustments in the MS were well justified and methodological uncertainties were added and discussed. The title has been updated and now reflects the main message of the paper adequately. From my perspective the paper is now suitable to be published in Nature Communications.

Reviewer #3 (Remarks to the Author):

I am satisfied with the revisions that the authors made to the manuscript through their extensive responses to my previous concerns; well done! I think the revised version has tied up some loose ends and I recommend this manuscript for publication.

Point-by-point response to the reviewers' comments

Reviewer #2 (Remarks to the Author):

Revision of manuscript "Recovery of logged forest fragments in a human-modified tropical landscape during the 2015-16 El Niño"

I would like to thank the authors for this very mindful and thorough revision of the manuscript. I believe that the main concerns raised by the reviewers were answered satisfactorily in much detail and changes well incorporated in the manuscript. Sections of the discussion and conclusion that were lacking some details were supplemented and accordingly improved. Adjustments in the MS were well justified and methodological uncertainties were added and discussed. The title has been updated and now reflects the main message of the paper adequately. From my perspective the paper is now suitable to be published in Nature Communications.

Thanks for the very thoughtful points raised by the reviewer.

Reviewer #3 (Remarks to the Author):

I am satisfied with the revisions that the authors made to the manuscript through their extensive responses to my previous concerns; well done! I think the revised version has tied up some loose ends and I recommend this manuscript for publication.

Thank you for the well-elaborated concerns that were crucial for the article's improvement.